# Properties and limitations of geometric tempering for gradient flow dynamics

**Francesca Romana Crucinio**                               *francescaromana.crucinio@unito.it*
*ESOMAS, University of Turin, Italy*
*Collegio Carlo Alberto, Turin, Italy*

**Sahani Pathiraja**                                              *s.pathiraja@unsw.edu.au*
*School of Mathematics & Statistics, UNSW Sydney, Australia*

**Reviewed on OpenReview:** *https://openreview.net/forum?id=IPOw5LdcxC&noteId=IOHESHSaVf*

## Abstract

We consider the problem of sampling from a probability distribution $\pi$. It is well known that this can be written as an optimisation problem over the space of probability distributions in which we aim to minimise the Kullback–Leibler divergence from $\pi$. We consider the effect of replacing $\pi$ with a sequence of moving targets $(\pi_t)_{t \geq 0}$ defined via geometric tempering on the Wasserstein and Fisher–Rao gradient flows. We show that convergence occurs exponentially in continuous time, providing novel bounds in both cases. We also consider popular time discretisations and explore their convergence properties. We show that in the Fisher–Rao case, replacing the target distribution with a geometric mixture of initial and target distribution never leads to a convergence speed up both in continuous time and in discrete time. Finally, we explore the gradient flow structure of tempered dynamics and derive novel adaptive tempering schedules.

## 1 Introduction

Sampling from a target probability distribution whose density is known up to a normalisation constant is a fundamental task in computational statistics and machine learning. A natural way to formulate this task is optimisation of a functional measuring the dissimilarity to the target probability distribution, typically the Kullback–Leibler (KL) divergence due to the fact that the KL does not require knowledge of the normalisation constant (Chen et al. (2023, Theorem 4.1) and Crucinio (2025)).

When the space over which the optimisation is carried out is the Wasserstein space, i.e. probability distributions with bounded second moments equipped with the Wasserstein-2 distance (Ambrosio et al., 2008), the resulting gradient descent scheme is referred to as Wasserstein (W) gradient flow.

A second class of gradient flows which has gained popularity in recent years is that of Fisher–Rao (FR) gradient flows, i.e. gradient descent w.r.t. the Fisher–Rao or Hellinger–Kakutani distance. Contrary to the Wasserstein metric, which leads to a diffusive behaviour, the Fisher–Rao metric leads to birth-death dynamics in which mass is created or removed (Lu et al., 2019; 2023).

Convergence of the W gradient flow is guaranteed when the target distribution is sufficiently smooth and verifies a log-Sobolev assumption (Vempala & Wibisono, 2019). Exponential convergence of the FR gradient flow for KL does not require a log-Sobolev assumption and can be established under a bounded second moment assumption and mild regularity of the ratio between target and initial distribution (Chen et al., 2023, Theorem 4.1 and Eq. (B.6)), and therefore holds for a wider class of targets.

Combinining the W and the FR gradient flows by making use of the Wasserstein–Fisher–Rao metric (Gallouët & Monsaingeon, 2017) leads to the Wasserstein–Fisher–Rao (WFR) gradient flow. The WFR flow combines the diffusive behaviour of the W gradient flow with the FR flow and enjoys more favourable convergence

properties, combining the rate of convergence of the W flow with that of the FR flow (Domingo-Enrich & Pooladian, 2023; Lu et al., 2019); its numerical approximation can considerably speed up the convergence of gradient based methods for multimodal targets (Lu et al., 2019; 2023).

In this work we consider *tempered* versions of the Wasserstein, Fisher–Rao and Wasserstein–Fisher–Rao PDEs obtained by considering a time varying target $\pi_t \propto \pi^{\lambda_t} \mu_0^{1-\lambda_t}$ for $\lambda_t : \mathbb{R}_+ \to [0,1]$. This is inspired by recent work that has considered similar tempered targets in methods that locally transport mass (Chehab et al., 2025; Máté & Fleuret, 2024; Guo et al., 2025a;b). Our focus is both on methods based on transport as the Wasserstein gradient flow and on methods based on reweighting and birth–death dynamics as the Fisher–Rao gradient flow. While it is known that the FR gradient flow of the KL is intimately tied to tempering or annealing due to its unit time rescaling (Domingo-Enrich & Pooladian, 2023; Chen et al., 2024), it is not immediately obvious to what extent adding further tempering schedules improves or degrades convergence.

We therefore study the continuous and discrete time convergence of the tempered W, FR and WFR flows (Section 2). We compare our results for the W flow with Chehab et al. (2025) and highlight similarities and improvements. In Section 3 we investigate the gradient flow structure of the tempered W and FR flow by jointly optimising tempering schedules and the evolution of the tempered flows. We compare the solution of the gradient flow approach with other approaches commonly used in the literature.

Our main contributions are:

- We provide novel results on the tempered W flow which clearly decompose the bound on the KL decay as the usual bound for the untempered W flow with a bias term which only disappears for $\lambda_t \equiv 1$ (Proposition 1 and 4).

- We show convergence of the tempered FR dynamics both in continuous and in discrete time (Proposition 2 and 5). We also show that combining tempering with the FR dynamics *never* improves the KL decay (Proposition 3).

- We explore the gradient flow structure of these tempered dynamics to obtain the steepest descent evolution for $\lambda$. This point of view allows us to compare the standard untempered flows with the tempered dynamics with the optimal tempering schedule. Theoretical and empirical results show that the steepest descent schedule does not improve over the untempered dynamics and is slower than other approaches commonly used in the literature (Section 3 and Section 4).

**Notation** We define some notation that will be used throughout the manuscript. We endow $\mathbb{R}^d$ with the Borel $\sigma$-field $\mathcal{B}(\mathbb{R}^d)$ with respect to the Euclidean norm $\|\cdot\|$ where $d$ is clear from context. We denote by $C^1(\mathbb{R}^d)$ the set of continuously differentiable functions, for all $f \in C^1(\mathbb{R}^d)$ we denote the gradient by $\nabla f$. Throughout this manuscript we denote the target by $\pi$ and assume $\pi = \exp(-V_\pi)$ for some $V_\pi \in C^1(\mathbb{R}^d)$. The initial distribution for the tempering iterates is denoted by $\mu_0 = \exp(-V_0)$ with $V_0 \in C^1(\mathbb{R}^d)$. Given $\pi, \mu_0$ we define $\pi_t \propto \pi^{\lambda_t} \mu_0^{1-\lambda_t}$ for $\lambda_t : \mathbb{R}_+ \to [0,1]$. We denote by $\mathcal{P}(\mathbb{R}^d)$ the set of probability measures over $\mathcal{B}(\mathbb{R}^d)$, and endow this space with the topology of weak convergence. For any $p \in \mathbb{N}$, we denote by $\mathcal{P}_p(\mathbb{R}^d) = \{\pi \in \mathcal{P}(\mathbb{R}^d) : \int_{\mathbb{R}^d} \|x\|^p d\pi(x) < +\infty\}$ the set of probability measures over $\mathcal{B}(\mathbb{R}^d)$ with finite $p$-th moment. We denote the subset of $\mathcal{P}(\mathbb{R}^d)$ of measures which are absolutely continuous w.r.t. Lebesgue by $\mathcal{P}^{ac}(\mathbb{R}^d)$ and let $\mathcal{P}_p^{ac}(\mathbb{R}^d) := \mathcal{P}_p(\mathbb{R}^d) \cap \mathcal{P}^{ac}(\mathbb{R}^d)$. For any $\mu, \nu \in \mathcal{P}_p(\mathbb{R}^d)$ we define the $p$-Wasserstein distance $W_p(\mu, \nu)$ between $\mu$ and $\nu$ by

$$W_p(\mu, \nu) = \left( \inf_{\gamma \in \mathbf{T}(\mu,\nu)} \int_{\mathbb{R}^d \times \mathbb{R}^d} \|x - y\|^p d\gamma(x,y) \right)^{1/p}$$

where $\mathbf{T}(\mu, \nu) = \{\gamma \in \mathcal{P}(\mathbb{R}^d \times \mathbb{R}^d) : \gamma(A \times \mathbb{R}^d) = \mu(A), \ \gamma(\mathbb{R}^d \times A) = \nu(A) \ \forall A \in \mathcal{B}(\mathbb{R}^d)\}$ denotes the set of all transport plans between $\mu$ and $\nu$. In the following, we metrise $\mathcal{P}_p(\mathbb{R}^d)$ with $W_p$.

## 2 Tempered PDEs for sampling

We consider a class of PDEs in which the target $\pi$ is replaced by a sequence of moving targets $(\pi_t)_{t \geq 0}$. We refer to these PDEs as *tempered* PDEs. The original non-tempered dynamics can be obtained by replacing

$\pi_t$ with $\pi$ throughout. Samplers based on tempered dynamics have become popular in the literature with the hope that the introduction of moving targets will improve the convergence of the sampler (Nüsken et al., 2024; Albergo & Vanden-Eijnden, 2024; Chehab et al., 2025; Guo et al., 2025a).

We focus here on the sequence $(\pi_t)_{t \geq 0}$ defined by the geometric tempering iteration $\pi_t \propto \pi^{\lambda_t} \mu_0^{1-\lambda_t}$ where the tempering schedule $\lambda_t : \mathbb{R}_+ \to [0, 1]$ is non-decreasing in $t$ and weakly differentiable. This has been popularised in the sampling literature through its use in sequential Monte Carlo (SMC; Del Moral et al. (2006)), annealed importance sampling (AIS; Neal (2001)), parallel tempering (PT; Geyer (1991)) and the homotopy method (Daum & Huang, 2008).

The tempering sequence $\lambda_t$ is fixed a priori. Popular options include the linear rate $\lambda_s = s/t$, the exponential rate $\lambda_s = 1 - e^{-\alpha s}$ for $\alpha > 0$ and the optimal rate for log-concave targets in Chehab et al. (2025, Section 4.2). We empirically compare this sequences in Section 4.

## 2.1 Tempered Wasserstein dynamics

We start by considering the tempered Wasserstein flow of Chehab et al. (2025). The corresponding PDE is given by

$$\partial_t \mu_t(x) = \nabla \cdot \left( \mu_t(x) \nabla \log \left( \frac{\mu_t(x)}{\pi_t(x)} \right) \right) \quad t \in [0, \infty). \tag{1}$$

This PDE is not a gradient flow of $D_{\mathrm{KL}}(\mu||\pi)$ and thus decay of $D_{\mathrm{KL}}(\mu||\pi)$ along the flow needs to be checked directly (similarly to e.g. Chehab et al. (2025, Theorem 1)).

We now analyse the decay of the Kullback–Leibler divergence along the tempered W PDE and compare it with its standard non-tempered counterpart. In order to obtain our results we assume that $\pi, \mu_0$ are strongly log-concave. These assumptions are strong but somewhat classical in the study of convergence of Langevin based samplers (e.g., Durmus et al. (2019); Chehab et al. (2025)).

**Assumption 1.** *The negative log-densities $V_\pi, V_0$ have Lipschitz continuous gradients with Lipschitz constants $L_\pi, L_0$, respectively. In addition, there exists $c_\pi, c_0 > 0$ such that*

$$\langle x - x', \nabla V_\pi(x) - \nabla V_\pi(x') \rangle \geq c_\pi \|x - x'\|^2, \qquad \langle x - x', \nabla V_0(x) - \nabla V_0(x') \rangle \geq c_0 \|x - x'\|^2.$$

Assumption 1 implies that $\pi$ satisfies a log-Sobolev inequality (e.g., Chewi et al. (2024)):

$$D_{\mathrm{KL}}(\mu||\pi) \leq \frac{C_{\mathrm{LSI}}}{2} \int_{\mathbb{R}^d} \mu(x) \|\nabla \log \frac{\mu(x)}{\pi(x)}\|^2 dx. \tag{2}$$

with log-Sobolev constant $C_{\mathrm{LSI}} = c_\pi^{-1}$. In addition, Assumption 1 implies the dissipativity condition

$$\langle x, \nabla V_\pi(x) \rangle \geq a_\pi \|x\|^2 - b_\pi,$$

for some $a_\pi > 0$ and $b_\pi > 0$. Assumption 1 also implies that $\nabla V_t := (1 - \lambda_t) \nabla V_0 + \lambda_t \nabla V_\pi$ is Lipschitz continuous with Lipschitz constant $L_t = (1 - \lambda_t) L_0 + \lambda_t L_\pi$ and strongly convex with constant $c_t \geq (1 - \lambda_t) c_0 + \lambda_t c_\pi$.

**Proposition 1** (Tempered W). *Under Assumption 1, the tempered W PDE equation 1 reduces the Kullback–Leibler divergence at rate*

$$D_{\mathrm{KL}}(\mu_t||\pi) \leq e^{-2tc_\pi} D_{\mathrm{KL}}(\mu_0||\pi) + A \int_0^t e^{-2(t-s)c_\pi}(1 - \lambda_s) ds, \tag{3}$$

*where $A < \infty$ is a constant which only depends on the dissipativity and Lipschitz properties of $V_\pi, V_0$. The convergence rate of the standard W PDE is recovered for $\lambda_s \equiv 1$.*

The proof of Proposition 1 can be found in Appendix B.1. The first term corresponds to the rate of the classical W PDE, to which a bias term is added due to the fact that $\pi_t$ approaches $\pi$ only as $t \to \infty$. The bias term only disappears for $\lambda_s \equiv 1$, corresponding to the standard Wasserstein gradient flow.

To allow comparison with Chehab et al. (2025, Theorem 1) we report below its statement (adapted to our assumptions):

**Theorem 1** (Chehab et al. (2025); Theorem 1). *Suppose Assumption 1 holds. Let $(c_t)_{t\geq0}$ be the convexity constant of $\pi_t$. Let $\mu_t$ be the law of equation 1 with initialization $p_0$ and denote by $\dot\lambda_t$ the weak time derivative of the tempering schedule. Then, for all $t \geq 0$:*

$$D_{\mathrm{KL}}(\mu_t||\pi) \leq \exp\left(-2\int_0^t c_s\, ds\right) D_{\mathrm{KL}}(p_0||\mu_0) + A\,(1-\lambda_t) + A\int_0^t \dot\lambda_s \exp\left(-2\int_s^t c_v\, dv\right) ds,$$

*where $p_0$ denotes the initial distribution of equation 1 and $A < \infty$ is a constant which only depends on the dissipativity and Lipschitz properties of $V_\pi, V_0$, the second moment of the initial distribution $p_0$, and linearly on the dimension $d$.*

The main difference is in the first term, which shows exponential decay: in our case this decay is driven by $c_\pi$ and the initial condition is $D_{\mathrm{KL}}(\mu_0||\pi)$, in the case of Chehab et al. (2025, Theorem 1) the exponential decay is driven by $c_s$ and the initial condition is $D_{\mathrm{KL}}(p_0||\mu_0)$. Thus, the first term in Chehab et al. (2025, Theorem 1) does not correspond to the standard KL decay result for the untempered W flow. This difference is due to the proof technique: instead of decomposing $D_{\mathrm{KL}}(\mu_t||\pi) = D_{\mathrm{KL}}(\mu_t||\pi_t) + \mathbb{E}_{\mu_t}[\log\mu_t/\pi]$ and then taking the time derivative $\frac{d}{dt}D_{\mathrm{KL}}(\mu_t||\pi)$ as in Chehab et al. (2025), we take the time derivative $\frac{d}{dt}D_{\mathrm{KL}}(\mu_t||\pi)$ and then decompose the result into a bias term and the relative Fisher information w.r.t. $\pi$.

We note that our upper bound in Proposition 1 is minimised for $\lambda_t \equiv 1$ contrary to Chehab et al. (2025, Corollary 5) which shows $\lambda_t \equiv 1$ minimises their upper bound only if $c_\pi \geq c_{\mu_0}$.

In addition, if $p_0 = \mu_0$, as it is often the case in practice, the result of Chehab et al. (2025, Theorem 1) reduces to

$$D_{\mathrm{KL}}(\mu_t||\pi) \leq A\,(1-\lambda_t) + A\int_0^t \dot\lambda_s \exp\left(-2\int_s^t c_v\, dv\right) ds,$$

and may not guarantee an exponential decay to zero as shown in the following example.

**Example 1.** *Let $\mu_0(x) = \mathcal{N}(x; 0, 1)$ and $\pi(x) = \mathcal{N}(x; m, 1)$ and consider the linear schedule $\lambda_s = s/t$ defined on $[0, t]$. Then $\pi_s(x) = \mathcal{N}(x; s/tm, 1)$ so that $c_s \equiv 1$. Then,*

$$\int_0^t \dot\lambda_s \exp\left(-2\int_s^t c_v\, dv\right) ds = \frac{1}{t}\int_0^t \exp(-2(t-2))\, ds = \frac{1 - e^{-2t}}{2t}$$

*which behaves like $1/(2t)$ as $t \to \infty$.*

## 2.2 Tempered Fisher–Rao dynamics

Given the tempering sequence $\pi_t$, one can also define the tempered FR PDE

$$\partial_t\mu_t(x) = \mu_t(x)\left(\log\left(\frac{\pi_t(x)}{\mu_t(x)}\right) - \mathbb{E}_{\mu_t}\left[\log\left(\frac{\pi_t}{\mu_t}\right)\right]\right). \tag{4}$$

Similarly to equation 1, the PDE equation 4 is not a gradient flow of $D_{\mathrm{KL}}(\mu||\pi)$. The tempered FR flow has analytic solution (see Appendix B.2 for a proof)

$$\mu_t \propto \mu_0^{e^{-t}+\int_0^t e^{s-t}(1-\lambda_s)ds}\pi^{\int_0^t e^{s-t}\lambda_s ds}. \tag{5}$$

We now show that equation 4 reduces $D_{\mathrm{KL}}(\mu||\pi)$. We consider the same assumptions required to obtain the convergence results for the standard FR flow in Chen et al. (2023, Theorem 4.1).

**Assumption 2.** *Assume that $\mu_0, \pi$ have bounded second moments $\int_{\mathbb{R}^d}\|x\|^2\mu_0(x)dx \leq B$, $\int_{\mathbb{R}^d}\|x\|^2\pi(x)dx \leq B$ and*

$$e^{-M(1+\|x\|^2)} \leq \frac{\mu_0(x)}{\pi(x)} \leq e^{M(1+\|x\|^2)} \tag{6}$$

*for some $M > 0$.*

Under this assumption, we can show that equation 4 reduces $D_{\mathrm{KL}}(\mu||\pi)$ at a rate which is always worse than that of the standard FR PDE (see Appendix B.3 for a proof).

**Proposition 2** (Tempered FR). *Suppose that Assumption 2 holds. The tempered FR PDE equation 4 reduces the Kullback–Leibler divergence at rate*

$$D_{\mathrm{KL}}(\mu_t||\pi) \le M(1 - \int_0^t e^{s-t}\lambda_s ds)\left[2 + B + B\exp\left(M(1+B)(1 - \int_0^t e^{s-t}\lambda_s ds)\right)\right]. \tag{7}$$

*The convergence result for the standard FR flow in Chen et al. (2023, Theorem 4.1) is recovered setting $\lambda_s \equiv 1$.*

Since $1 - \int_0^t e^{s-t}\lambda_s ds - e^{-t} = \int_0^t e^{s-t}(1 - \lambda_s)ds \ge 0$, we conclude that the tempered FR PDE does not improve on the convergence rate of the standard FR PDE given by $Me^{-t}\left[2 + B + B\exp\left(M(1+B)e^{-t}\right)\right]$.

Comparing equation 5 with the evolution along the standard FR PDE (see, e.g., Chen et al. (2023))

$$\mu_t^{\mathrm{FR}} \propto \mu_0^{e^{-t}} \pi^{1-e^{-t}}, \tag{8}$$

it is easy to observe that both evolutions belong to the family of geometric interpolations

$$\rho_\alpha(x) := \frac{\mu_0(x)^{1-\alpha}\pi(x)^\alpha}{Z(\alpha)}, \quad Z_\alpha := \int \mu_0(x)^{1-\alpha}\pi(x)^\alpha \, dx, \quad \alpha \in [0,1]. \tag{9}$$

This observation allows us to quantify the difference in KL between the tempered FR (T-FR) and standard FR flows (see Appendix B.4 for a proof).

**Proposition 3.** *Let $\mu_t^{T\text{-}FR}$ denote the evolution of the tempered FR flow equation 5 and $\mu_t^{FR}$ the evolution of the standard FR flow equation 8. Then*

$$D_{\mathrm{KL}}(\mu_t^{FR}||\pi) - D_{\mathrm{KL}}(\mu_t^{T\text{-}FR}||\pi) = -\int_{\int_0^t e^{s-t}\lambda_s ds}^{1-e^{-t}}(1-u)\operatorname{Var}_{\rho_u}\left(\log\frac{\mu_0}{\pi}\right)du \le 0,$$

*for all $t \ge 0$.*

This shows that the standard FR flow is always faster than the tempered FR flow, the difference in speed depends on the choice of $\lambda_s$ and on the ratio $\mu_0/\pi$.

### 2.3 Tempered Wasserstein–Fisher–Rao dynamics

Given the tempered W and tempered FR PDE one can naturally define a tempered WFR PDE

$$\partial_t\mu_t = \nabla \cdot \left(\mu_t(x)\nabla\log\left(\frac{\mu_t(x)}{\pi_t(x)}\right)\right) + \mu_t(x)\left(\log\left(\frac{\pi_t(x)}{\mu_t(x)}\right) - \mathbb{E}_{\mu_t}\left[\log\left(\frac{\pi_t}{\mu_t}\right)\right]\right). \tag{10}$$

By defining the derivative of $D_{\mathrm{KL}}(\mu||\pi)$ along the tempered W flow equation 1 and that along the tempered FR flow equation 4 as

$$\frac{d}{dt}D_{\mathrm{KL}}(\mu_t^{\mathrm{T\text{-}W}}||\pi) := -\int_{\mathbb{R}^d}\mu_t\|\nabla\log\frac{\mu_t}{\pi}\|^2 + (1-\lambda_t)\int_{\mathbb{R}^d}\log\frac{\mu_t}{\pi}\nabla\cdot\left(\mu_t\nabla\log\left(\frac{\pi}{\mu_0}\right)\right)$$

$$\frac{d}{dt}D_{\mathrm{KL}}(\mu_t^{\mathrm{T\text{-}FR}}||\pi) := -\operatorname{Var}_{\mu_t}\left(\log\frac{\mu_t}{\pi}\right) - (1-\lambda_t)\operatorname{Cov}_{\mu_t}\left(\log\frac{\mu_t}{\pi}, \log\frac{\pi}{\mu_0}\right)$$

we can write

$$\frac{d}{dt}D_{\mathrm{KL}}(\mu_t^{\mathrm{T\text{-}WFR}}||\pi) = \frac{d}{dt}D_{\mathrm{KL}}(\mu_t^{\mathrm{T\text{-}W}}||\pi) + \frac{d}{dt}D_{\mathrm{KL}}(\mu_t^{\mathrm{T\text{-}FR}}||\pi).$$

This is directly comparable to the property that the WFR metric approximately decomposes as the sum of the W and FR metric as shown in (Gallouët & Monsaingeon, 2017).

Exploiting the fact that the time derivative of $D_{\mathrm{KL}}$ along the tempered WFR PDE factorises into two terms as in the case of the standard WFR flow, if $\mu_t$ follows the tempered WFR flow equation 10,

$$D_{\mathrm{KL}}(\mu_t|\pi) \leq \min\left\{D_{\mathrm{KL}}(\mu_t^{\mathrm{T\text{-}W}}||\pi), D_{\mathrm{KL}}(\mu_t^{\mathrm{T\text{-}FR}}||\pi)\right\}; \tag{11}$$

where an upper bound to $D_{\mathrm{KL}}(\mu_t^{\mathrm{T\text{-}W}}||\pi)$ is given in equation 3 and one for $D_{\mathrm{KL}}(\mu_t^{\mathrm{T\text{-}FR}}||\pi)$ in equation 7.

### 2.4 Multivariate Gaussian case

Our results as well as those in Chehab et al. (2025) suggest that the tempered dynamics do not improve on the gradient flow ones discussed in the previous section. However, our convergence results only obtain upper bounds for the decay of $D_{\mathrm{KL}}$ which are not necessarily tight. To showcase that the tempered PDEs do not improve over the standard ones, we consider the special case of transporting a Gaussian $\mu_0(x) = \mathcal{N}(x; m_0, \Sigma_0)$ to a target Gaussian $\pi(x) = \mathcal{N}(x; m_\pi, \Sigma_\pi)$, for which the W, FR, WFR, and their tempered versions can be solved analytically (see Appendix C).

In particular, observing that both the tempered W flow and the tempered FR flow preserve gaussianity, we have that $\mu_t(x) = \mathcal{N}(x; m_t, \Sigma_t)$ for each time $t \geq 0$ for equation 1, equation 4 and equation 10. The evolution of $m_t$ and $\Sigma_t$ is given by the following ODEs for the tempered W flow

$$\dot{m}_t = -(\lambda_t \Sigma_\pi^{-1} + (1-\lambda_t)\Sigma_0^{-1})\left(m_t - ((1-\lambda_t)m_0\Sigma_\pi + \lambda_t m_\pi \Sigma_0)(\lambda_t \Sigma_0 + (1-\lambda_t)\Sigma_\pi)^{-1}\right)$$
$$\dot{\Sigma}_t = -(\lambda_t \Sigma_\pi^{-1} + (1-\lambda_t)\Sigma_0^{-1})\Sigma_t - \Sigma_t(\lambda_t \Sigma_\pi^{-1} + (1-\lambda_t)\Sigma_0^{-1}) + 2I,$$

and for the tempered FR flow by

$$\dot{m}_t = -\Sigma_t(\lambda_t \Sigma_\pi^{-1} + (1-\lambda_t)\Sigma_0^{-1})(m_t - m_\pi)$$
$$\dot{\Sigma}_t = -\Sigma_t(\lambda_t \Sigma_\pi^{-1} + (1-\lambda_t)\Sigma_0^{-1})\Sigma_t + \Sigma_t.$$

The standard (i.e. non-tempered) flows are obtained by setting $\lambda_t \equiv 1$ in the above equations. The tempered WFR flow is obtained by summing the r.h.s. of the W and the FR flows.

Figure 1 shows the evolution of mean, covariance and Kullback–Leibler divergence along different PDE flows in the case of 1D Gaussians. For the tempered PDEs we use the linear schedule $\lambda_s = s/t$. The initial distribution is a standard Gaussian and we consider two different targets: for target 1 (first row) we set $m_\pi = 20, \Sigma_\pi = 0.1$. In this case the W flow is faster than the FR one and the WFR flow behaves more like the W flow (Figure 1 top row). For the second target (second row) we have $m_\pi = 1, \Sigma_\pi = 5$; the FR flow is faster than the W flow for all sufficiently large $t$, and the WFR flow more closely follows the FR flow. Figure 1 confirms our theoretical results: the tempered PDEs result in a slower convergence to the target than the corresponding standard flows.

### 2.5 Discrete Time Convergence

The previous sections suggest that tempering does not improve the continuous time dynamics given by the W, FR and WFR flows. In this section, we consider time discretisations of the W and FR dynamics and show that, also in this case, tempering does not allow us to obtain sharper decay bounds for the W flow and decelerates in convergence of the FR flow.

#### 2.5.1 Tempered Unadjusted Langevin Algorithm

Tempered unadjusted Langevin algorithms (Tempered ULA) Chehab et al. (2025) provide an approximation of the tempered Wasserstein PDE equation 1 by using an Euler discretisation of the associated SDE

$$dX_t = \nabla \log \pi_t(X_t)dt + \sqrt{2}dB_t. \tag{12}$$

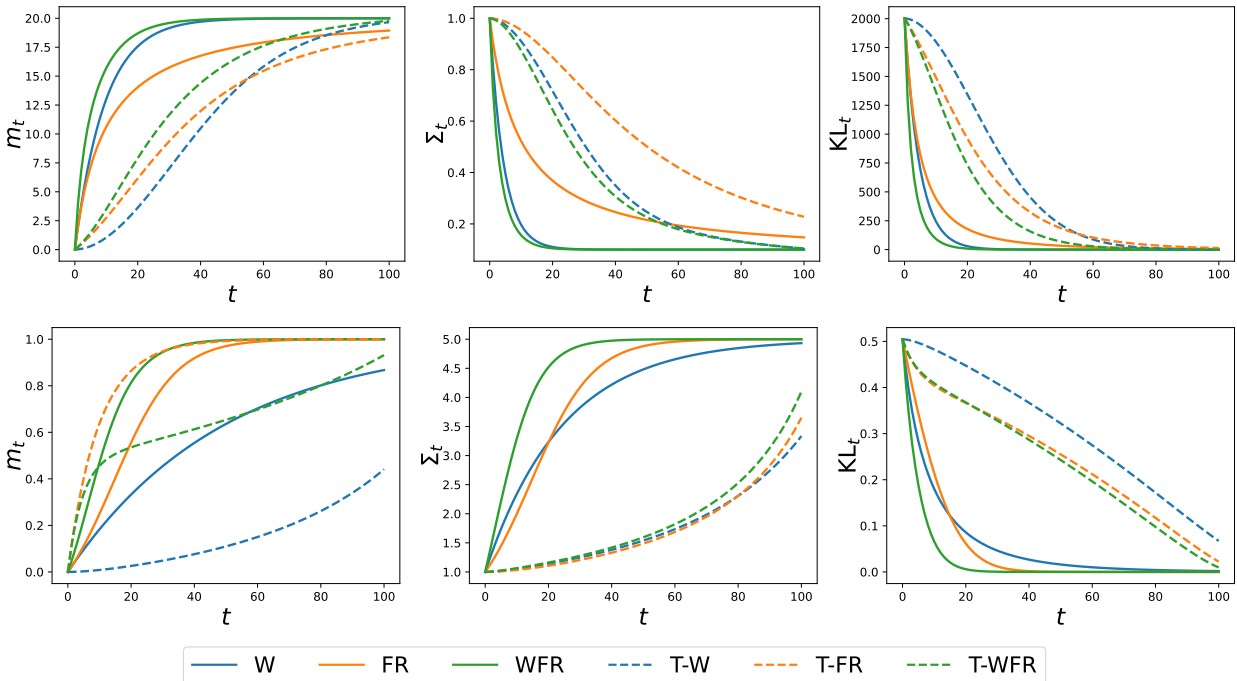

Figure 1: Evolution of mean, variance and $D_{\mathrm{KL}}(\mu_t || \pi)$ with $\mu_t$ given by the PDE flows in equation 1, equation 4 and equation 10 in the 1D Gaussian case with $\mu_0(x) = \mathcal{N}(x; 0, 1)$ and $\pi(x) = \mathcal{N}(x; 20, 0.1)$ (first row), $\pi(x) = \mathcal{N}(x; 1, 5)$ (second row).

Denote by $\pi_n \propto \pi^{\lambda_n} \mu_0^{1-\lambda_n}$ obtained by considering a time discretisation $0 = \lambda_0 \leq \lambda_1 \leq \cdots \leq \lambda_n \leq \lambda_{n+1} \leq \ldots$ of $\lambda_t : \mathbb{R}_+ \to [0, 1]$ with $\lambda_n \equiv \lambda_{t_n}$. The Tempered ULA dynamics for discretisation steps $(\gamma_n)_{n \geq 0}$ are given by

$$X_n = X_{n-1} + \gamma_n \nabla \log \pi_n(X_{n-1}) + \sqrt{2\gamma_n} \xi_n, \tag{13}$$

where $\xi_n \sim \mathcal{N}(0, \mathsf{Id})$. Denoting by $\mu_{n-1}$ the law of $X_{n-1}$, the update equation 13 corresponds to

$$\mu_n(x) = (4\pi\gamma_n)^{-d/2} \int_{\mathbb{R}^d} \exp\left( -\frac{1}{4\gamma_n} \|x - y - \gamma_n \nabla \log \pi_n(y)\|^2 \right) \mu_{n-1}(y) dy$$

where $\mu_n$ denotes the law of $X_n$.

Under Assumption 1 we can then obtain an equivalent result to Proposition 1 in discrete time, decomposing the KL decay into the classical rate for standard W flows and a bias term (see Appendix D.1 for a proof).

**Proposition 4.** *Under Assumption 1, if $\gamma_n \leq \min(1, \frac{a_\pi \wedge a_0}{2(L_\pi + L_0)^2}, \frac{c_n}{4L_n^2})$, then*

$$D_{\mathrm{KL}}(\mu_n || \pi) \leq D_{\mathrm{KL}}(\mu_0 || \pi) e^{-\sum_{k=1}^n \gamma_k c_k} + \sum_{k=0}^{n-1} (6\gamma_k^2 d L_k^2 + (1 - \lambda_k) A') e^{-\sum_{i=1}^k \gamma_i c_i},$$

*where $A'$ depends on the constants from Assumption 1, the second moment of the initial distribution $\mu_0$, and linearly on the dimension.*

To allow comparison with Chehab et al. (2025, Theorem 3) we report below its statement (adapted to our assumptions):

**Theorem 2** (Chehab et al. (2025); Theorem 3). *Suppose Assumption 1 holds. Let $(c_k)_{k \geq 0}$ be the convexity constant of $\pi_k$. Let $X_0 \sim p_0$ and $X_n$ be given by equation 13. Then, if $\gamma_n \leq \min(1, \frac{a_\pi \wedge a_0}{2(L_\pi + L_0)^2}, \frac{c_n}{4L_n^2})$,*

$$D_{\mathrm{KL}}(\mu_n || \pi) \leq D_{\mathrm{KL}}(p_0 || \mu_0) e^{-\sum_{k=1}^n \gamma_k c_k} + 6 \sum_{k=1}^n \gamma_k^2 d L_k^2 e^{-\sum_{i=k+1}^n \gamma_i c_i}$$

$$+ A'(1 - \lambda_k) + A' \sum_{i=1}^n (\lambda_i - \lambda_{i-1}) \exp\left(-2 \sum_{j=i}^k \gamma_j c_j\right)$$

*where $A'$ depends on the constants from Assumption 1, the second moment of the initial distribution $p_0$, and linearly on the dimension.*

Also in discrete time case, our result differs from Chehab et al. (2025, Theorem 3). In Proposition 4 the first two terms correspond to exponential decay driven by $c_\pi$ with initial condition $D_{\mathrm{KL}}(\mu_0 || \pi)$ and a time discretisation bias which is the rate of standard ULA (e.g. Vempala & Wibisono (2019, Theorem 1)); the following terms are the bias which disappears if $\lambda_k \equiv 1$. This is still the case when $p_0 = \mu_0$, contrary to Chehab et al. (2025, Theorem 3).

### 2.5.2 Tempered Entropic Mirror Descent

Consider the tempered FR PDE equation 4; instead of using the exact evolution equation 5 consider an explicit time discretisation with discretisation steps $(\gamma_n)_{n \geq 0}$

$$\log(\mu_n) - \log(\mu_{n-1}) = -\gamma_n \left(\log\left(\frac{\mu_{n-1}}{\pi_{n-1}}\right) - \mathbb{E}_{\mu_{n-1}}\left[\log\left(\frac{\mu_{n-1}}{\pi_{n-1}}\right)\right]\right), \tag{14}$$

where $\pi_n \propto \pi^{\lambda_n} \mu_0^{1-\lambda_n}$ Observing that $\mathbb{E}_{\mu_{n-1}}\left[\log\left(\frac{\mu_{n-1}}{\pi_{n-1}}\right)\right]$ is just a normalising constant, we obtain the sequence of distributions

$$\mu_n \propto \mu_{n-1}^{(1-\gamma_n)} \pi_{n-1}^{\gamma_n} = \mu_{n-1}^{(1-\gamma_n)} \pi^{\lambda_{n-1}\gamma_n} \mu_0^{(1-\lambda_{n-1})\gamma_n}. \tag{15}$$

Proceeding recursively the above becomes

$$\mu_n \propto \mu_0^{1 - \sum_{k=1}^n \gamma_k \lambda_{k-1} \prod_{j=k+1}^n (1-\gamma_j)} \pi^{\sum_{k=1}^n \gamma_k \lambda_{k-1} \prod_{j=k+1}^n (1-\gamma_j)},$$

showing that a time discretisation of the tempered FR PDE belongs to the family of geometric tempering equation 9 with $\alpha_n = \sum_{k=1}^n \gamma_k \lambda_{k-1} \prod_{j=k+1}^n (1-\gamma_j)$.

We point out that, as for the classical FR flow, this time discretisation leads to a discrete time sequence of distributions but not to a practical algorithm. In Section 4 we consider an importance based numerical approximation of the (tempered) FR flow.

Observing that equation 15 corresponds to one step of entropic mirror descent with time step $\alpha_n = \sum_{k=1}^n \gamma_k \lambda_{k-1} \prod_{j=k+1}^n (1 - \gamma_j)$ applied to the Kullback–Leibler divergence (see Chopin et al. (2024)), allows us to obtain the following result, which is a direct corollary of Chopin et al. (2024, Proposition 1) with $\gamma_n$ replaced by $\alpha_n$ therein.

**Proposition 5.** *Let $\alpha_n = \sum_{k=1}^n \gamma_k \lambda_{k-1} \prod_{j=k+1}^n (1 - \gamma_j)$. Then the time discretisation of the tempered FR flow equation 15 satisfies*

$$D_{\mathrm{KL}}(\mu_n || \pi) \leq (\alpha_1)^{-1} \prod_{k=1}^n (1 - \alpha_k) D_{\mathrm{KL}}(\pi || \mu_0).$$

It follows that, since $\alpha_n \leq 1$ for all $n \geq 1$, the tempered FR flow reduces KL at a geometric rate $\prod_{k=1}^n (1 - \alpha_k) \to 0$ as $n \to \infty$.

To compare the tempered FR dynamics with the standard FR dynamics we recall that the discretisation of the standard FR flow

$$\mu_n \propto \mu_0^{\prod_{k=1}^n (1-\gamma_k)} \pi^{1-\prod_{k=1}^n (1-\gamma_k)}$$

belongs to the family of geometric tempering equation 9 with $\tilde{\alpha}_n = 1 - \prod_{k=1}^n (1-\gamma_k)$. Using that

$$\tilde{\alpha}_n = 1 - \prod_{k=1}^n (1-\gamma_k) = \sum_{k=1}^n \gamma_k \prod_{j=k+1}^n (1-\gamma_j) \geq \sum_{k=1}^n \gamma_k \lambda_{k-1} \prod_{j=k+1}^n (1-\gamma_j) = \alpha_n,$$

and the fact that if $\alpha_1 > \alpha_2$ then (see Appendix B.4 for a proof) $D_{\mathrm{KL}}(\rho_{\alpha_1}||\pi) - D_{\mathrm{KL}}(\rho_{\alpha_2}||\pi) < 0$, we conclude that the discretisation of the tempered FR PDE at iteration $n$ has larger KL divergence from $\pi$ than the discretisation of the standard FR PDE.

## 3 Gradient flow structure of tempered PDEs

The tempered PDEs described above do not correspond to gradient flows of $D_{\mathrm{KL}}(\mu||\pi)$. To identify a gradient flow structure, we consider the functional $\mathcal{F}(\mu, \lambda) : \mathcal{P}(\mathbb{R}^d) \times [0,1] \to \mathbb{R}$ defined as $\mathcal{F}(\mu, \lambda) := D_{\mathrm{KL}}(\mu||\pi_\lambda) + D_{\mathrm{KL}}(\pi_\lambda||\pi)$ where $\pi_\lambda \propto \mu_0^{1-\lambda}\pi^\lambda$. The unique minimiser of $\mathcal{F}(\mu, \lambda)$ is $(\mu, \lambda) = (\pi, 1)$ for which $\mathcal{F}^\star = \mathcal{F}(\pi, 1) = 0$. Thus $\mathcal{F}(\mu, \lambda)$ admits the same global minimiser of $D_{\mathrm{KL}}(\mu||\pi)$.

A gradient flow of $\mathcal{F}(\mu, \lambda)$ optimises over both $\mu$ and $\lambda$, leading to a sequence $\lambda_t$ which is not fixed a priori but learnt as time progresses. Our aim is to understand if the sequence $(\lambda_t)_{t \geq 0}$ adaptively chosen using this gradient flow point of view can lead to dynamics which converge faster than those obtained with classical tempering strategies and to investigate whether it can beat the untempered dynamics.

### 3.1 Tempered Fisher–Rao dynamics

To interpret the tempered Fisher–Rao dynamics as a gradient flow, we consider the functional $\mathcal{F}(\lambda, \mu)$ and the Fisher–Rao metric on $\mathcal{P}(\mathbb{R}^d)$ coupled with the Euclidean metric on $[0, 1]$.

The Fisher–Rao gradient flow of $\mathcal{F}$ is given by Carrillo et al. (2024)

$$\partial_t \mu_t(x) = -\mu_t(x) \left( \frac{\delta \mathcal{F}(\mu_t, \lambda)}{\delta \mu_t}(x) - \mathbb{E}_{\mu_t} \left[ \frac{\delta \mathcal{F}(\mu_t, \lambda)}{\delta \mu_t} \right] \right) = -\mu_t(x) \left( \log \frac{\mu_t(x)}{\pi_{\lambda_t}(x)} - \mathbb{E}_{\mu_t} \left[ \log \frac{\mu_t(x)}{\pi_{\lambda_t}(x)} \right] \right) \quad (16)$$

$\frac{\delta \mathcal{F}(\mu, \lambda)}{\delta \mu}$ denotes the first variation of $\mathcal{F}$ at w.r.t. $\mu$ for fixed $\lambda$ (see Appendix E for a derivation), which clearly coincides with the tempered FR flow. The gradient flow of $\lambda_t$ is given again by

$$\dot{\lambda}_t = \int_{\mathbb{R}^d} \mu_t(x) \log \frac{\pi(x)}{\mu_0(x)} dx - \int_{\mathbb{R}^d} \pi_{\lambda_t}(x) \log \frac{\pi(x)}{\mu_0(x)} dx + (1 - \lambda_t) \mathrm{Var}_{\pi_{\lambda_t}} \left( \log \frac{\mu_0}{\pi} \right). \quad (17)$$

The next result shows that $\mathcal{F}$ has a unique minimiser at $(\pi, 1)$ and that this point is asymptotically stable, so that $(\mu_t, \lambda_t) \to (\pi, 1)$ as $t \to \infty$. The proof can be found in Appendix E.2.

**Proposition 6.** *Let $\mathcal{F}(\mu, \lambda) := D_{\mathrm{KL}}(\mu||\pi_\lambda) + D_{\mathrm{KL}}(\pi_\lambda||\pi)$. Consider the gradient flow dynamics equation 16 and equation 17. Then the unique minimiser $(\pi, 1)$ is asymptotically stable and thus $(\mu_t, \lambda_t) \to (\pi, 1)$ as $t \to \infty$.*

In the case of the tempered FR flow, Proposition 3 guarantees that the fastest KL decay is obtained with $\lambda_t \equiv 1$. We thus check on an example if the optimal schedule identified in equation 17 quickly approaches the constant function $\lambda_t \equiv 1$.

**Example 2.** *Let $\mu_0(x) = \mathcal{N}(x; 0, 1)$ and $\pi(x) = \mathcal{N}(x; m, 1)$. Then $\pi_\lambda(x) = \mathcal{N}(x; \lambda m, 1)$, $\log \frac{\mu_0(x)}{\pi(x)} = mx - m^2/2$ and*

$$Var_{\pi_{\lambda_t}} \left( \log \frac{\mu_0}{\pi} \right) = m^2.$$

*We also have that $\mu_t(x) = \mathcal{N}(x; m_t, \Sigma_t)$ with $m_t, \Sigma_t$ given in Appendix C. We thus have*

$$\dot{\lambda}_t = m \int_{\mathbb{R}^d} [\pi_{\lambda_t}(x) - \mu_t(x)] x dx + (1 - \lambda_t) m^2 = m^2 (1 - \int_0^t e^{s-t} \lambda_s ds)$$

*as $m_t = m \int_0^t e^{s-t} \lambda_s ds$. The solution of this ODE with initial condition $\lambda_0 = 0$ is*

$$\lambda_t = 1 - \frac{m^2 + r_2}{r_2 - r_1} e^{r_1 t} - \frac{m^2 + r_1}{r_1 - r_2} e^{r_2 t}, \qquad r_{1,2} = \frac{-1 \pm \sqrt{1 - 4m^2}}{2} \tag{18}$$

*which only has a solution in $\mathbb{R}$ if $m^2 \leq 1/4$. Taking the limit $t \to \infty$ we find that $\lambda_t \to 1$. Figure 2 shows the evolution of $\lambda_t$ for $m = 1/6$. As $r_2 < r_1 < 0$, $\lambda_t \sim 1 - \frac{m^2 + r_2}{r_2 - r_1} e^{r_1 t}$ as $t \to \infty$ and thus the rate of convergence is given by $e^{r_1 t}$. This rate of convergence is the fastest when $m^2 \approx 1/4$ for which*

$$\lambda_t = 1 - \left(1 - \frac{t}{2}\right) e^{-t/2},$$

*showing that the gradient flow of $\mathcal{F}(\lambda, \mu)$ w.r.t. $\lambda$ approaches 1 at most at rate $te^{-t/2}$.*

## 3.2 Tempered Wasserstein dynamics

To define a gradient flow of $\mathcal{F}$ we need to consider a metric on $\mathcal{P}(\mathbb{R}^d) \times [0, 1]$. To derive the tempered Wasserstein dynamics equation 1 from a gradient flow point of view, we consider the Wasserstein metric on $\mathcal{P}(\mathbb{R}^d)$ coupled with the Euclidean metric on $[0, 1]$. The metric structure of this product space is investigated by Kuntz et al. (2023, Appendix A), which guarantees that $\mathcal{P}(\mathbb{R}^d) \times [0, 1]$ is a metric space.

We follow Kuntz et al. (2023) and consider $\nabla \mathcal{F}(\mu, \lambda) = (\nabla_\mu \mathcal{F}(\mu, \lambda), \partial_\lambda \mathcal{F}(\mu, \lambda))$, where $\nabla_\mu$ denotes the gradient w.r.t. $\mu$ in Wasserstein distance and $\partial_\lambda$ the usual derivative in Euclidean space. Using Kuntz et al. (2023, Lemma 1 and Eq. (18)), we find

$$\nabla_\mu \mathcal{F}(\mu, \lambda) = \nabla \cdot \left(\mu \nabla \log \frac{\pi_\lambda}{\mu}\right),$$

$$\partial_\lambda \mathcal{F}(\mu, \lambda) = -\int_{\mathbb{R}^d} \mu(x) \partial_\lambda \log \pi_\lambda(x) dx + \partial_\lambda D_{\mathrm{KL}}(\pi_\lambda || \pi).$$

Observing that (see Appendix B.4 for a proof)

$$\partial_\lambda \log \pi_\lambda(x) = \log \frac{\pi(x)}{\mu_0(x)} - \int_{\mathbb{R}^d} \pi_\lambda(x) \log \frac{\pi(x)}{\mu_0(x)} dx$$

$$\partial_\lambda D_{\mathrm{KL}}(\pi_\lambda || \pi) = -(1 - \lambda) \mathrm{Var}_{\pi_\lambda}\left(\log \frac{\mu_0}{\pi}\right),$$

we obtain the gradient flow dynamics

$$\partial_t \mu_t(x) = -\nabla \cdot \left(\mu_t(x) \nabla \log \frac{\pi_{\lambda_t}(x)}{\mu_t(x)}\right)$$

$$\dot{\lambda}_t = \int_{\mathbb{R}^d} \mu_t(x) \log \frac{\pi(x)}{\mu_0(x)} dx - \int_{\mathbb{R}^d} \pi_{\lambda_t}(x) \log \frac{\pi(x)}{\mu_0(x)} dx + (1 - \lambda_t) \mathrm{Var}_{\pi_{\lambda_t}}\left(\log \frac{\mu_0}{\pi}\right) \tag{19}$$

It is now easy to see that the evolution of $\mu_t$ coincides with the tempered W PDE equation 1 provided that the appropriate $\lambda_t$ is chosen.

The decay of $\mathcal{F}$ along these dynamics is given by

$$\frac{d}{dt} \mathcal{F}(\lambda_t, \mu_t) = \frac{d}{dt} D_{\mathrm{KL}}(\mu_t || \pi_{\lambda_t}) + D_{\mathrm{KL}}(\pi_{\lambda_t} || \pi)$$

$$= \int_{\mathbb{R}^d} \log \frac{\mu_t(x)}{\pi_{\lambda_t}(x)} \partial_t \mu_t(x) dx - \int_{\mathbb{R}^d} \mu_t(x) \partial_t \log \pi_{\lambda_t}(x) dx + \frac{d}{d\lambda_t} D_{\mathrm{KL}}(\pi_{\lambda_t} || \pi) \dot{\lambda}_t.$$

Using the derivative of KL w.r.t. $\lambda$ derived in Appendix B.4, the fact that

$$\partial_\lambda \log \pi_\lambda(x) = \log \frac{\pi(x)}{\mu_0(x)} - \int_{\mathbb{R}^d} \pi_\lambda(x) \log \frac{\pi(x)}{\mu_0(x)} dx,$$

and integration by parts, we obtain

$$\frac{d}{dt} \mathcal{F}(\lambda_t, \mu_t) = - \int_{\mathbb{R}^d} \|\nabla \log \frac{\mu_t(x)}{\pi_{\lambda_t}(x)}\|^2 \mu_t(x) dx - \dot{\lambda}_t \int_{\mathbb{R}^d} [\mu_t(x) - \pi_{\lambda_t}(x)] \log \frac{\pi(x)}{\mu_0(x)} dx - (1 - \lambda_t) \mathrm{Var}_{\pi_{\lambda_t}} \left[ \log \frac{\mu_0}{\pi} \right] \dot{\lambda}_t \tag{20}$$

$$= - \int_{\mathbb{R}^d} \|\nabla \log \frac{\mu_t(x)}{\pi_{\lambda_t}(x)}\|^2 \mu_t(x) dx - (\dot{\lambda}_t)^2, \tag{21}$$

confirming that the rate of decay of $\mathcal{F}$ is driven both by the tempered W PDE and the optimally chosen $\lambda_t$. With this, we obtain an equivalent result to Proposition 6 in the tempered Wasserstein setting; the proof can be found in Appendix E.3.

**Proposition 7.** *Let $\mathcal{F}(\mu, \lambda) := D_{\mathrm{KL}}(\mu||\pi_\lambda) + D_{\mathrm{KL}}(\pi_\lambda||\pi)$. Consider the gradient flow dynamics equation 19. Then the unique minimiser $(\pi, 1)$ is asymptotically stable and thus $(\mu_t, \lambda_t) \to (\pi, 1)$ as $t \to \infty$.*

In Section 4.4 we empirically compare the tempered W dynamics given by equation 19 with other popular adaptive choices of $\lambda_t$.

### 3.3 Comparison with Popular Tempering Schedules

We now compare popular tempering schedules in the sequential Monte Carlo (SMC) and Annealed Importance Sampling (AIS) literature with equation 17. To connect with popular adaptive schedules in the literature, let us consider the KL decay along the sequence $(\pi_{\lambda_t})_{t \geq 0}$ given by

$$\frac{d}{dt} D_{\mathrm{KL}}(\pi_{\lambda_t}||\pi) = -(1 - \lambda_t) \mathrm{Var}_{\pi_{\lambda_t}} \left[ \log \frac{\mu_0}{\pi} \right] \dot{\lambda}_t.$$

A natural choice to adaptively select $\lambda_t$ is to require a constant KL decay, i.e. $\frac{d}{dt} D_{\mathrm{KL}}(\mu_t||\pi) = -\beta$ for $\beta > 0$ (Goshtasbpour et al., 2023). Solving the ODE

$$\dot{\lambda}_t = \beta \left( (1 - \lambda_t) \mathrm{Var}_{\pi_{\lambda_t}} \left[ \log \frac{\mu_0}{\pi} \right] \right)^{-1}$$

leads to the adaptive tempering schedule proposed in Goshtasbpour et al. (2023).

Consider now the exact solution of the standard FR gradient flow equation 8, which is obtained from $\pi_{\lambda_t}$ if $\lambda_t = 1 - e^{-t}$. In this case $\frac{d\lambda_t}{dt} = 1 - \lambda_t$ and

$$\frac{d}{dt} D_{\mathrm{KL}}(\pi_{\lambda_t}||\pi) = -(\dot{\lambda}_t)^2 \mathrm{Var}_{\pi_{\lambda_t}} \left( \log \frac{\mu_0}{\pi} \right) = \beta. \tag{22}$$

The resulting ODE

$$\dot{\lambda}_t = \sqrt{\beta} \mathrm{Var}_{\pi_{\lambda_t}} \left( \log \frac{\mu_0}{\pi} \right)^{-1/2} \tag{23}$$

gives an adaptive strategy which corresponds to the classical strategy to adaptively set $\lambda_t$ in SMC and AIS by controlling the $\chi^2$ divergence between successive iterates (Chopin et al., 2024).

To compare the three adaptive schedules equation 22, equation 23 and equation 17 we consider two examples.

**Example 3.** *Let $\mu_0(x) = \mathcal{N}(x; 0, 1)$ and $\pi(x) = \mathcal{N}(x; m, 1)$. In this case, the evolution along equation 17 with initial condition $\lambda_0 = 0$ is given by equation 18, while the solutions of equation 22, equation 23 are given by*

$$\lambda_t = 1 - \sqrt{1 - \frac{2t}{m^2}}, \qquad t \leq \frac{m^2}{2} \qquad \qquad \text{for equation 22}$$

$$\lambda_t = mt \qquad \qquad \text{for equation 23.}$$

*Figure 2 (left) compares the evolutions above for $m = 1/6$.*

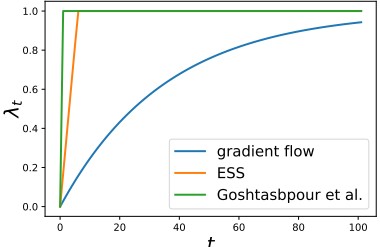 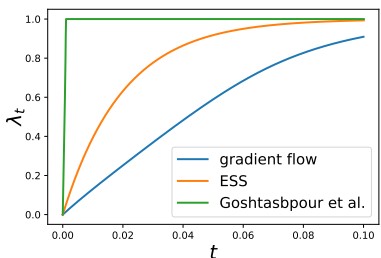 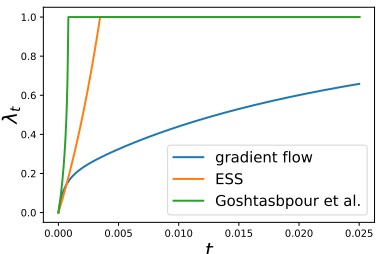

Figure 2: Comparison of tempering sequences for tempered dynamics with $\mu_0(x) = \mathcal{N}(x; 0, 1)$ and $\pi(x) = \mathcal{N}(x; m, 1)$ with $m = 1/6$ (left), $\pi(x) = \mathcal{N}(x; 0, \tau^2)$ with $\tau^2 = 2$ (middle) and $\pi(x) = \mathcal{N}(x; 0, \tau^2)$ with $\tau^2 = 0.5$ (right).

**Example 4.** *Let* $\mu_0(x) = \mathcal{N}(x; 0, 1)$ *and* $\pi(x) = \mathcal{N}(x; 0, \tau^2)$. *Then* $\pi_\lambda(x) = \mathcal{N}\left(x; 0, \left(1 - \lambda + \lambda/\tau^2\right)^{-1}\right)$ *and*

$$\log \frac{\mu_0(x)}{\pi(x)} = \log \tau + \frac{x^2}{2}\left(\frac{1}{\tau^2} - 1\right)$$

$$Var_{\pi_{\lambda_t}}\left(\log \frac{\mu_0}{\pi}\right) = \frac{1}{2}\left(\frac{1}{\tau^2} - 1\right)^2 \left(1 - \lambda + \lambda/\tau^2\right)^{-2}.$$

*The evolution with initial condition* $\lambda_0 = 0$ *can only be solved analytically for equation 23 and is given by* $\lambda_t = 1 - \exp([1/\tau^2 - 1]t)$ *if* $\tau > 1$ *and by* $\lambda_t = \exp([1/\tau^2 - 1]t) - 1$ *if* $\tau < 1$. *In this case, equation 17 reduces to*

$$\dot{\lambda}_t = \frac{1}{2}\left(\frac{1}{\tau^2} - 1\right)\left(\frac{1}{(1 - \lambda) + \lambda/\tau^2} - \frac{1}{1 - \int_0^t e^{s-t}\lambda_s ds + \int_0^t e^{s-t}\lambda_s ds/\tau^2}\right)$$

$$+ (1 - \lambda_t)\frac{1}{2}\left(\frac{1}{\tau^2} - 1\right)^2 \left(1 - \lambda + \lambda/\tau^2\right)^{-2}$$

*as* $\mu_t(x) = \mathcal{N}(x; 0, \left(1 - \int_0^t e^{s-t}\lambda_s ds + \int_0^t e^{s-t}\lambda_s ds/\tau^2\right)^{-1})$. *Figure 2 (middle and right) compares the evolutions above where equation 22 and equation 17 are solved numerically.*

In all cases, the schedule corresponding to the gradient flow equation 17 is slower than classical schedules found in the literature. We believe this is due to the fact that to identify a gradient flow structure for the tempered PDEs we consider the functional $\mathcal{F}(\mu, \lambda) := D_{\mathrm{KL}}(\mu||\pi_\lambda) + D_{\mathrm{KL}}(\pi_\lambda||\pi)$, while commonly used approaches can be linked to the time derivative of $D_{\mathrm{KL}}(\mu_t||\pi)$ when $\mu_t$ follows the FR gradient flow.

We observe that, in practice, equation 22, equation 23 and equation 17 are intractable, as they require computing an expectation against $\pi_{\lambda_t} \propto \pi^{\lambda_t}\mu_0^{1-\lambda_t}$ which is known up to a normalising constant. In the AIS and SMC literature, it is usually assumed that the samples that track the evolution of $\pi_{\lambda_t}$ provide a good approximation and thus use $\mu_t$ as an approximation of $\pi_{\lambda_t}$. As $\mu_t$ closely tracks $\pi_{\lambda_t}$ by construction in our case, this is reasonable to assume $\mu_t \approx \pi_{\lambda_t}$ and obtain

$$\dot{\lambda}_t = (1 - \lambda_t)\mathrm{Var}_{\pi_{\lambda_t}}\left(\log \frac{\mu_0}{\pi}\right) \approx (1 - \lambda_t)\mathrm{Var}_{\mu_t}\left(\log \frac{\mu_0}{\pi}\right),$$

providing a computable form for the steepest descent ODE for $\lambda_t$. We test this in Section 4.4.

## 4 Numerical Experiments

We now present some numerical experiments to support our results in the previous sections. Code to reproduce our experiments will be made available. Code to reproduce our experiments is available at `https://github.com/FrancescaCrucinio/SMC-WFR`.

### 4.1 Algorithms

We briefly describe here the numerical approximations of the tempered W PDE and the tempered WFR PDE that we use in the following sections.

For the tempered W PDE we use the Tempered ULA dynamics equation 13. For the tempered WFR PDE we adapt the sequential Monte Carlo algorithm of Crucinio & Pathiraja (2025a) to the use of tempered targets.

To obtain a time discretisation of the tempered WFR PDE we consider the solution operator of the tempered W flow and the tempered FR flow separately. Assume that for certain discrete time $n-1$ the solution of the tempered WFR PDE is given by $\mu_{n-1}$. Given $\mu_{n-1}$, a discretisation of the tempered W flow for a time step $\gamma_n$ leads to

$$\mu_{n-1/2}(x) \propto \int_{\mathbb{R}^d} \exp\left(-\frac{1}{4\gamma_n}\|x - y - \gamma_n \nabla \log \pi_n(y)\|^2\right) \mu_{n-1}(y)dy,$$

which can be implemented using the tempered ULA equation 13. Using the exact solution of the tempered FR PDE equation 5 we have

$$\mu_n(x) \propto \mu_{n-1/2}(x)^{e^{-\gamma}+\int_0^\gamma e^{s-\gamma}(1-\lambda_s)ds} \pi(x)^{\int_0^\gamma e^{s-\gamma}\lambda_s ds} = \left(\frac{\pi(x)}{\mu_{n-1/2}(x)}\right)^{\int_0^\gamma e^{s-\gamma}\lambda_s ds} \mu_{n-1/2}(x), \qquad (24)$$

where we used $e^{-t} + \int_0^t e^{s-t}(1-\lambda_s)ds = 1 - \int_0^t e^{s-t}\lambda_s ds$. Given $\mu_{n-1/2}$, we can obtain $\mu_n$ by importance sampling with weights $\left(\pi(x)/\mu_{n-1/2}(x)\right)^{\int_0^\gamma e^{s-\gamma}\lambda_s ds}$.

Alternating between one step of tempered W flow approximated via equation 13 and one step of tempered FR flow approximated via importance sampling we obtain an approximation of tempered WFR PDE through sampling-based procedures. A resampling step to avoid the weight degeneracy normally introduced by repeated importance sampling steps can also be added (Algorithm 1).

---

**Algorithm 1** SMC-T-WFR.

---

1: *Inputs:* learning rates $(\gamma_n)_{n\geq 0}$, initial proposal $\mu_0$.
2: *Initialize:* sample $\widetilde{X}_0^i \sim \mu_0$ and set $W_0^i = 1/N$ for $i = 1, \ldots, N$.
3: **for** $n = 1, \ldots, T$ **do**
4:     **if** $n > 1$ **then**
5:       *Resample:* draw $\{\widetilde{X}_{n-1}^i\}_{i=1}^N$ independently from $\{X_{n-1}^i, W_{n-1}^i\}_{i=1}^N$ and set $W_n^i = 1/N$ for $i = 1, \ldots, N$.
6:     **end if**
7:     *W flow:* update $X_n^i = \bar{X}_{n-1}^i + \sqrt{2\gamma_n}\xi_{n-1/2}^i$, where $\bar{X}_{n-1}^i = \widetilde{X}_{n-1}^i + \gamma_n \nabla \log \pi(\widetilde{X}_{n-1}^i)$, $\xi_{n-1/2}^i \sim \mathcal{N}(0, \mathsf{Id})$ for $i = 1, \ldots, N$.
8:     *FR flow:* compute and normalize the weights $W_n^i \propto w_n(X_n^i)$ for $i = 1, \ldots, N$, where

$$w_n(x) = \left(\frac{\pi(x)}{N^{-1}\sum_{i=1} \mathcal{N}(x; X_{n-1}^i + \gamma_n \nabla \log \pi_n(X_{n-1}^i), 2\gamma_n \cdot \mathsf{Id})}\right)^{\int_0^{\gamma_n} e^{s-\gamma_n}\lambda_s ds}. \qquad (25)$$

9: **end for**
10: *Output:* $\{X_n^i, W_n^i\}_{i=1}^N$.

---

Despite using the tempered unadjusted Langevin algorithm as proposal, Algorithm 1 does not coincide with the standard tempering SMC sampler (Chopin & Papaspiliopoulos, 2020, Chapter 17). To draw a connection between the two, consider the time discretisation of the tempered FR PDE in equation 15 with $\gamma_n \equiv 1$, $\mu_n \propto \pi^{\lambda_n}\mu_0^{(1-\lambda_n)}$. Using this sequence of distributions with the tempered unadjusted Langevin proposal equation 13 coincides with the commonly used tempering SMC and results in weights $w_n(x_n) = (\pi(x_n)/\mu_0(x_n))^{\lambda_n - \lambda_{n-1}}$ for sufficiently small $\gamma$ (Dai et al., 2022).

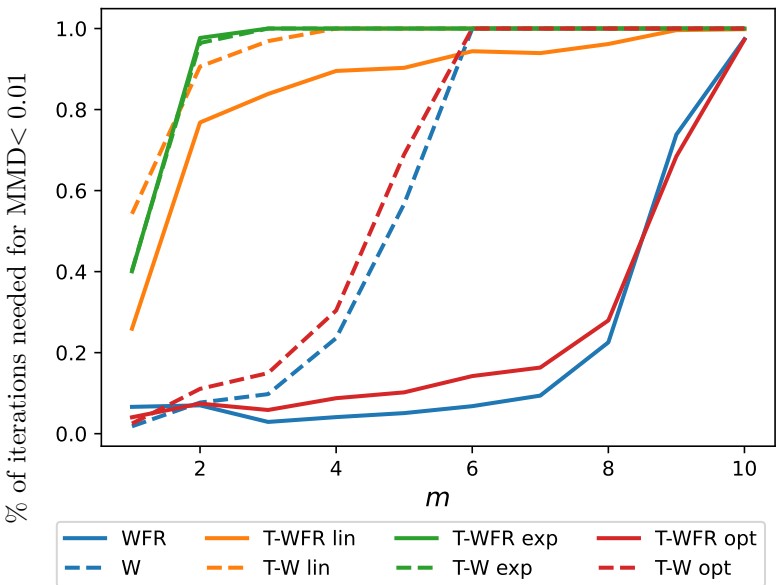

Figure 3: Convergence of approximations to $\pi$ given by tempered and standard PDE flow with target the Gaussian mixture equation 26. Convergence is measured through the MMD with standard Gaussian kernel and is averaged over 50 replications. We plot the percent of iterations required to achieve MMD$< 0.01$ for increasing $m$. We include standard W & WFR and tempered flows with $\lambda_s = s/t$, $\lambda_s = 1 - e^{-\alpha s}$, $\lambda_s = 1 - 1/(2 + s)$.

## 4.2 Tempered and Standard PDEs

Proposition 1 shows that the tempered Wasserstein dynamics (T-W) do not improve on the standard Wasserstein gradient flows (W) in the case of strongly log-concave target $\pi$. A similar result holds for the WFR flow combining the rates in Proposition 1 with those in Proposition 2 as shown in equation 11. For bi-modal targets, which satisfy a log-Sobolev assumption but are not log-concave, (Chehab et al., 2025, Theorem 8) shows that the T-W flow has exponentially slower convergence than the standard W flow.

We empirically show that the same is true for the tempered WFR flow. We consider the same setup of Chehab et al. (2025):

$$\mu_0(x) = \mathcal{N}(x; 0, 1), \qquad \pi(x) = \frac{1}{2}\mathcal{N}(x; 0, 1) + \frac{1}{2}\mathcal{N}(x; m, 1). \qquad (26)$$

This target satisfies a log-Sobolev inequality (Schlichting, 2019) with $C_{\text{LSI}} = (1 + (e^{m^2} + 1)/2)$.

For the tempered PDEs we consider the linear rate $\lambda_s = s/t$, the exponential rate $\lambda_s = 1 - e^{-\alpha s}$ for $\alpha = 0.01$ and the optimal rate of Chehab et al. (2025), $\lambda_s = 1 - 1/(2 + s)$ (which follows from setting $\mu_0(x) = \mathcal{N}(x; 0, 1)$). We check convergence computing the MMD with standard Gaussian kernel.

Figure 3 shows the number of iteration needed to achieve MMD$< 0.01$. For the W flows we run $N$ parallel chains using the same number of iterations and discretisation steps as for the WFR flows. We see that the tempered dynamics do not perform better than the standard dynamics; the optimal rate performs similarly to the non-tempered dynamics for both W and WFR. WFR converges faster than the W flow for all choices of tempering regardless of the log-Sobolev constant $C_{\text{LSI}}$. The increased convergence speed of WFR comes at a higher computational cost; while ULA and its tempered variants have $\mathcal{O}(N)$ cost, WFR and Tempered WFR require the computation of the weights leading to an $\mathcal{O}(N^2)$ cost (Crucinio & Pathiraja, 2025a).

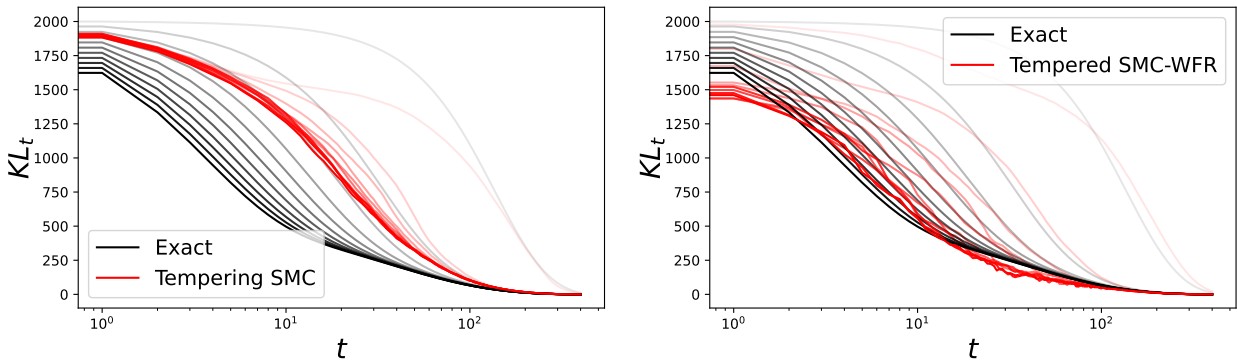

Figure 4: Evolution of $D_{\mathrm{KL}}$ along the tempered WFR PDE flow with linear schedule and two approximations via SMC. We have $\mu_0(x) = \mathcal{N}(x; 0, 1)$ and $\pi(x) = \mathcal{N}(x; 20, 0.1)$, $\gamma$ ranges from 0.001 (lighter color) to 0.1 (darker color).

### 4.3 Tempered SMC-WFR and Tempering SMC

We now compare two approximations of the tempered WFR flow: Algorithm 1 obtained using the analytic solution of the FR flow and the usual tempering SMC algorithm (e.g. Chopin & Papaspiliopoulos (2020)) which uses the sequence $\mu_n \propto \pi^{\lambda_n} \mu_0^{(1-\lambda_n)}$ and tempered ULA equation 13 as proposal kernel.

We empirically compare this two approximations on a 1D Gaussian example with $\mu_0(x) = \mathcal{N}(x; 0, 1)$ and $\pi(x) = \mathcal{N}(x; 20, 0.1)$ using the linear schedule $\lambda_s = s/t$. We take $\gamma \in [0.001, 0.1]$, $N = 400$ and investigate how well the two approaches match the exact evolution of $D_{\mathrm{KL}}$ along the tempered WFR flow obtained using the moment ODEs in Appendix C. Figure 4 shows that for small $\gamma$ the two approximations are both close to the exact evolution along the tempered WFR PDE but as $\gamma$ increases tempering SMC deviates from the continuous time evolution by slowing down convergence.

Despite the slower convergence, tempering SMC has some advantages over the approximation of the tempered WFR flow given by Algorithm 1: setting $\gamma_n \equiv 1$ in equation 15 results in weights which only involve $\pi, \mu_0$ and thus can be computed in $\mathcal{O}(1)$ time against the $\mathcal{O}(N)$ time of equation 25. In addition, when using the sequence of distributions $\mu_n \propto \pi^{\lambda_n} \mu_0^{1-\lambda_n}$, one can set $\lambda_T = 1$ for some large $T$ and therefore obtain $\pi$ in finite time; this is not the case for the exact solution of the tempered FR PDE equation 5 for which convergence to $\pi$ only occurs in the infinite time limit.

Figure 5 shows the distribution of the effective sample size (ESS)

$$\mathrm{ESS}_n := \left( \sum_{i=1}^{N} (W_n^i)^2 \right)^{-1}$$

over iterations for tempering SMC and tempered SMC-WFR. We observe that tempering SMC guarantees a relative ESS always above 75% while for tempered SMC-WFR the relative ESS quickly degrades with increasing $\gamma$. For sufficiently small $\gamma$ (i.e. $\gamma = 0.001$) tempered SMC-WFR has higher ESS than tempering SMC.

### 4.4 Comparison of Tempering Schedules

We compare the adaptive tempering schedules in Section 3 on the target equation 26 with $m = 2$= We consider the tempered W dynamics in equation 1 and use $\gamma = 0.01$, $N = 500$ and $T = 500$ iterations.

Figure 6 shows the evolution of MMD and $\lambda_t$ for ULA and the three adaptive strategies. In both cases the gradient flow based ODE is the slowest to converge to 1 and thus the decay of MMD to 0 is the slowest. The strategy of Goshtasbpour et al. (2023) quickly converges to 1 and more closely follows the MMD decay

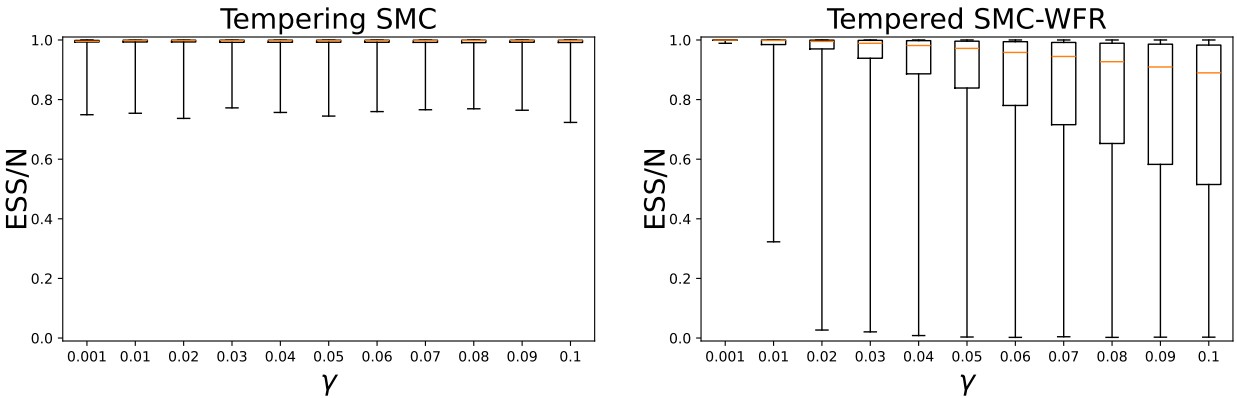

Figure 5: Distribution of ESS for tempered SMC-WFR and tempering SMC for different values of $\gamma$. We have $\mu_0(x) = \mathcal{N}(x; 0, 1)$ and $\pi(x) = \mathcal{N}(x; 20, 0.1)$, $\gamma$ ranges from 0.001 to 0.1.

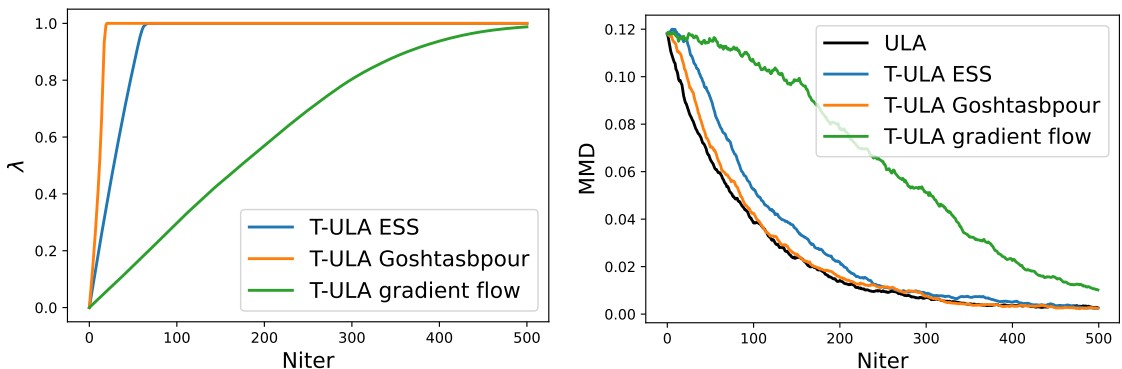

Figure 6: Comparison of adaptive strategies for $\lambda_t$ and resulting MMD decay for $\mu_0(x) = \mathcal{N}(x; 0, 1)$ and $\pi(x) = \frac{1}{2}\mathcal{N}(x; 0, 1) + \frac{1}{2}\mathcal{N}(x; m, 1)$.

of ULA. The ESS based strategy is slighlty slower. However, ULA remains the fastest strategy in terms of convergence to $\pi$.

## 5 Discussion

In this work, we provided a systematic analysis of geometric tempering in gradient flow dynamics, with a focus on the Fisher–Rao and Wasserstein settings. Our results offer a clearer understanding of the role and limitations of tempering in these frameworks.

In the Fisher–Rao case, we derived an explicit comparison between tempered and untempered dynamics, showing that tempering does not accelerate convergence and, in fact, leads to slower dynamics (Proposition 3). This conclusion holds in both continuous and discrete time and follows from an exact characterisation rather than a bound, highlighting a fundamental limitation of tempering in reweighting-based flows.

In the Wasserstein setting, we showed that the effect of tempering can be understood through a decomposition of the KL decay into a standard convergence term and an additional bias induced by the evolving target distribution (Proposition 1 and Proposition 4). This perspective clarifies how tempering modifies the dynamics and distinguishes our analysis from the work of Chehab et al. (2025).

We also investigated the construction of tempering schedules from a gradient flow perspective. While this provides a principled framework, both our theoretical and empirical findings indicate that the resulting schedules are typically slower than commonly used alternatives (Section 3 and Section 4.4).

Even if our theoretical and empirical results (Section 4) suggest that tempered dynamics are not beneficial for sampling, we point out that the idea of tempering, i.e. to slowly move the initial distribution $\mu_0$ to the target $\pi$ via a geometric path, coincides with the FR flow as highlighted in equation 9. It is then intuitive that tempering the FR flow itself is will not improve on its convergence as shown in Proposition 3.

For the W flow we argue that a more promising way to combine transport dynamics with tempering ideas is to consider Wasserstein–Fisher–Rao flows. In fact, these flows are provably faster than W and FR flows alone (Crucinio & Pathiraja, 2025b; Lu et al., 2023; 2019). We hope that this work contributes to a more precise understanding of the tempering phenomena and informs the design of future sampling methods.

### Acknowledgments

F.R.C. gratefully acknowledges the "de Castro" Statistics Initative at the *Collegio Carlo Alberto* and the *Fondazione Franca e Diego de Castro*. F.R.C. is supported by the Gruppo Nazionale per l'Analisi Matematica, la Probabilità e le loro Applicazioni (GNAMPA-INdAM).

S.P. gratefully acknowledges funding from UNSW Faculty of Science Research Grant and the Eva Mayr Stihl Foundation.

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

## A  Preliminary results

We collect here a number of preliminary results from Chehab et al. (2025) and Vempala & Wibisono (2019) which we will use in the proof of Proposition 1 and Proposition 4.

We recall that Assumption 1 implies the dissipativity condition

$$\langle x, \nabla V_\pi(x)\rangle \geq a_\pi \|x\|^2 - b_\pi. \tag{27}$$

To see this use Assumption 1 and the Cauchy-Schwarz inequality to obtain

$$\langle \nabla V_\pi(x), x\rangle = \langle \nabla V_\pi(x) - \nabla V_\pi(0), x\rangle + \langle \nabla V_\pi(0), x\rangle \geq c_\pi \|x\|^2 - \|x\|\|\nabla V_\pi(0)\|.$$

Then Young's inequality yields, $\alpha\beta \leq \alpha^2/(2\epsilon) + \beta^2\epsilon/2$ for all $\epsilon > 0$, and thus that:

$$-\|x\|\|\nabla V_\pi(0)\| \geq -\|x\|^2/(2\epsilon) - \epsilon\|\nabla V_\pi(0)\|^2/2.$$

It follows that for all $\epsilon > 1/(2c_\pi)$ we have equation 27 with $a_\pi = c_\pi - 1/(2\epsilon) > 0$ and $b_\pi = \epsilon\|\nabla V_\pi(0)\|^2/2 > 0$; and similarly for $\mu_0$.

We now list some results from Chehab et al. (2025) for tempered Langevin dynamics.

**Lemma 1** (Chehab et al. (2025); Lemma 15). *Let $\mu_t$ be the solution of equation 1. Under Assumption 1, for all $t \geq 0$,*

$$\mathbb{E}_{\mu_t}\left[\|x\|^2\right] \leq \max\left(\mathbb{E}_{\mu_0}\left[\|x\|^2\right], \frac{d + b_{\mu_0} + b_\pi}{a_{\mu_0} \wedge a_\pi}\right).$$

**Lemma 2** (Chehab et al. (2025); Proposition 14). *Under Assumption 1, for all $x \in \mathbb{R}^d$, we have*

$$V_{\mu_0}(x) - \inf_{y \in \mathbb{R}^d} V_{\mu_0}(y) \leq 2L_{\mu_0}\left(\|x\|^2 + \frac{b_{\mu_0}}{a_{\mu_0}}\right), \qquad V_\pi(x) - \inf_{y \in \mathbb{R}^d} V_\pi(y) \leq 2L_\pi\left(\|x\|^2 + \frac{b_\pi}{a_\pi}\right).$$

**Lemma 3** (Chehab et al. (2025); Lemma 16). *Suppose*

$$\gamma_n \leq \min\left(1, \frac{a_\pi \wedge a_{\mu_0}}{2(L_\pi + L_{\mu_0})^2}\right).$$

*Let $\mu_n$ be the distribution of tempered ULA equation 13 at time $n$. Then, under Assumption 1,*

$$\mathbb{E}_{\mu_n}\left[\|x\|^2\right] \leq \max\left(\mathbb{E}_{\mu_0}\left[\|x\|^2\right], \frac{2(3(b_\pi + b_{\mu_0})/2 + d)}{a_\pi \wedge a_{\mu_0} \wedge 1}\right).$$

We also include a result on convergence of ULA from Vempala & Wibisono (2019).

**Lemma 4** (Vempala & Wibisono (2019); Lemma 3). *Suppose $\nu$ satisfies a log-Sobolev inequality equation 2 with constant $C_{LSI} > 0$ and is $L$-smooth. If $0 < \gamma_n \leq \frac{C_{LSI}^{-1}}{4L^2}$, then along each step of ULA targeting $\nu \in \mathcal{P}(\mathbb{R}^d)$*

$$X_n = X_{n-1} + \gamma_n \nabla \log \nu(X_{n-1}) + \sqrt{2\gamma_n}\xi_n$$

*we have*

$$D_{\mathrm{KL}}(\mu_n\|\nu) \leq e^{-C_{LSI}^{-1}\gamma_n} D_{\mathrm{KL}}(\mu_{n-1}\|\nu) + 6\gamma_n^2 dL^2,$$

*where $(\mu_n)_{n\geq 0}$ denotes the distribution of $X_n$.*

# B  Continuous time convergence

## B.1  Proof of Proposition 1

A direct computation of the decay of $D_{\mathrm{KL}}(\mu_t||\pi)$ along this PDE gives

$$\frac{d}{dt}D_{\mathrm{KL}}(\mu_t||\pi) = \int_{\mathbb{R}^d} \log\frac{\mu_t(x)}{\pi(x)}\partial_t\mu_t(x)dx + \int_{\mathbb{R}^d} \partial_t\left(\log\frac{\mu_t(x)}{\pi(x)}\right)\mu_t(x)dx$$

$$= \int_{\mathbb{R}^d} \log\frac{\mu_t(x)}{\pi(x)}\partial_t\mu_t(x)dx + \int_{\mathbb{R}^d} \partial_t\mu_t(x)dx$$

$$= \int_{\mathbb{R}^d} \log\frac{\mu_t(x)}{\pi(x)}\partial_t\mu_t(x)dx$$

where we may interchange differentiation and integration given Assumption 1.

Consider the W PDE

$$\partial_t\mu_t(x) = \nabla\cdot\left(\mu_t(x)\nabla\log\left(\frac{\mu_t(x)}{\pi_t(x)}\right)\right)$$

where $\pi_t \propto \pi^{\lambda_t}\mu_0^{1-\lambda_t}$ and $\lambda_t : \mathbb{R}_+ \to [0,1]$, $\lambda_t$ is non-decreasing in $t$ and weakly differentiable. An application of integration by parts and the regularity guaranteed by Assumption 1 give

$$\frac{d}{dt}D_{\mathrm{KL}}(\mu_t||\pi) = -\int_{\mathbb{R}^d}\mu_t(x)\|\nabla\log\frac{\mu_t(x)}{\pi(x)}\|^2dx + (1-\lambda_t)\int_{\mathbb{R}^d}\log\frac{\mu_t(x)}{\pi(x)}\nabla\cdot\left(\mu_t(x)\nabla\log\left(\frac{\pi(x)}{\mu_0(x)}\right)\right)dx$$

$$= -\int_{\mathbb{R}^d}\mu_t(x)\|\nabla\log\frac{\mu_t(x)}{\pi(x)}\|^2dx$$

$$+ (1-\lambda_t)\int_{\mathbb{R}^d}\mu_t(x)\left[\Delta V_0(x) - \Delta V_\pi(x) - \langle\nabla V_\pi(x), \nabla V_0(x) - \nabla V_\pi(x)\rangle\right]dx$$

where we used the expression for $\pi_t$ and multiple applications of integrations by parts to obtain the last equality.

A consequence of the dissipativity equation 27 is

$$\mathbb{E}_\pi[\|X\|^2] \le \frac{1}{a_\pi}\mathbb{E}_\pi[\langle X, \nabla V_\pi(X)\rangle] + \frac{b_\pi}{a_\pi} = \frac{d+b_\pi}{a_\pi}. \tag{28}$$

In addition, $V_\pi$ must be minimised at some point $z_0 \in B(0, \sqrt{\frac{b_\pi}{a_\pi}})$, thus the Lipschitz assumption implies

$$\|\nabla V_\pi(z)\|^2 \le 2L_\pi^2\left(\|z\|^2 + \frac{b_\pi}{a_\pi}\right). \tag{29}$$

The same properties follow for $\mu_0$.

Using the above and Lemma 1 we have

$$\int_{\mathbb{R}^d}\mu_t(x)\left[\Delta V_0(x) - \Delta V_\pi(x) - \langle\nabla V_\pi(x), \nabla V_0(x) - \nabla V_\pi(x)\rangle\right]dx$$

$$\le d(L_{\mu_0} - c_\pi) + 2L_\pi^2\left(\mathbb{E}_{\mu_t}[\|x\|^2] + \frac{b_\pi}{a_\pi}\right) + \int_{\mathbb{R}^d}\mu_t(x)(\|\nabla V_\pi(x)\|^2 \vee \|\nabla V_0(x)\|^2)dx$$

$$\le d(L_{\mu_0} - c_\pi) + 4(L_\pi^2 \vee L_{\mu_0}^2)\left(\mathbb{E}_{\mu_t}[\|x\|^2] + \left(\frac{b_\pi}{a_\pi} \vee \frac{b_0}{a_0}\right)\right)$$

$$\le d(L_{\mu_0} - c_\pi) + 4(L_\pi^2 \vee L_{\mu_0}^2)\left(\mathbb{E}_{\mu_0}[\|x\|^2] \vee \frac{d + b_{\mu_0} + b_\pi}{a_{\mu_0} \wedge a_\pi} + \left(\frac{b_\pi}{a_\pi} \vee \frac{b_0}{a_0}\right)\right) := A.$$

Using Grönwall's Lemma we obtain

$$D_{\mathrm{KL}}(\mu_t||\pi) \le e^{-2tc_\pi}D_{\mathrm{KL}}(\mu_0||\pi) + A\int_0^t e^{-2(t-s)c_\pi}(1-\lambda_s)ds.$$

## B.2   Proof of equation 5

From equation 4 we find

$$\frac{\partial \log \mu_t(x)}{\partial t} = \log\left(\frac{\pi_t(x)}{\mu_t(x)}\right) - \mathbb{E}_{\mu_t}\left[\log\left(\frac{\pi_t}{\mu_t}\right)\right].$$

It follows that

$$\frac{\partial e^t \log \mu_t(x)}{\partial t} = e^t \frac{\partial \log \mu_t(x)}{\partial t} + e^t \log \mu_t(x)$$

$$= e^t \log \pi_t(x) - e^t \mathbb{E}_{\mu_t}\left[\log\left(\frac{\pi_t}{\mu_t}\right)\right].$$

Integrating both sides over $[0, t]$ we obtain

$$e^t \log \mu_t(x) - \log \mu_0(x) = \int_0^t \left(e^s \log \pi_s(x) - e^s \mathbb{E}_{\mu_s}\left[\log\left(\frac{\pi_s}{\mu_s}\right)\right]\right) ds,$$

or, equivalently,

$$\log \mu_t(x) = e^{-t} \log \mu_0(x) + \int_0^t \left(e^{s-t} \log \pi_s(x) - e^{s-t} \mathbb{E}_{\mu_s}\left[\log\left(\frac{\pi_s}{\mu_s}\right)\right]\right) ds,$$

Recall that $\pi_t \propto \mu_0^{1-\lambda_t} \pi^{\lambda_t}$, therefore

$$\log \mu_t(x) = \log \mu_0(x)(\int_0^t e^{s-t}(1-\lambda_s)ds + e^{-t}) + \log \pi(x) \int_0^t e^{s-t}\lambda_s ds - \int_0^t e^{s-t}\mathbb{E}_{\mu_s}\left[\log\left(\frac{\pi_s}{\mu_s}\right)\right] ds.$$

It follows that

$$\mu_t \propto \mu_0^{e^{-t}+\int_0^t e^{s-t}(1-\lambda_s)ds} \pi^{\int_0^t e^{s-t}\lambda_s ds}.$$

## B.3   Proof of Proposition 2

Consider the FR PDE

$$\partial_t \mu_t(x) = \mu_t(x)\left(\log\left(\frac{\pi_t(x)}{\mu_t(x)}\right) - \mathbb{E}_{\mu_t}\left[\log\left(\frac{\pi_t}{\mu_t}\right)\right]\right),$$

where $\pi_t \propto \pi^{\lambda_t} \mu_0^{1-\lambda_t}$ and $\lambda_t : \mathbb{R}_+ \to [0, 1]$, $\lambda_t$ is non-decreasing in $t$ and weakly differentiable. Using an argument similar to that of Chen et al. (2023, Appendix B.1) we can show that this PDE reduces the Kullback–Leibler divergence w.r.t. $\pi$.

The normalising constant of $\mu_t$ satisfies

$$Z_t := \int_{\mathbb{R}^d} \mu_0(x)^{1-\int_0^t e^{s-t}\lambda_s ds} \pi(x)^{\int_0^t e^{s-t}\lambda_s ds} dx$$

$$= \int_{\mathbb{R}^d} \pi(x)\left(\frac{\pi(x)}{\mu_0(x)}\right)^{1-\int_0^t e^{s-t}\lambda_s ds} dx$$

$$\geq \int_{\mathbb{R}^d} \pi(x) \exp\left(-M(1+\|x\|^2)(1-\int_0^t e^{s-t}\lambda_s ds)\right) dx$$

$$\geq \exp\left(\int_{\mathbb{R}^d} \pi(x)\left(-M(1+\|x\|^2)(1-\int_0^t e^{s-t}\lambda_s ds)\right) dx\right)$$

$$\geq \exp\left(-M(1+B)(1-\int_0^t e^{s-t}\lambda_s ds)\right),$$

where we used equation 6 and Jensen's inequality.

Since $\int_0^t e^{s-t}\lambda_s ds < 1$ we also have

$$
\begin{aligned}
\frac{\mu_t(x)}{\pi(x)} &= \frac{1}{Z_t} \frac{\mu_0(x)^{1-\int_0^t e^{s-t}\lambda_s ds} \pi(x)^{\int_0^t e^{s-t}\lambda_s ds}}{\pi(x)} \\
&= \frac{1}{Z_t} \left( \frac{\mu_0(x)}{\pi(x)} \right)^{1-\int_0^t e^{s-t}\lambda_s ds} \\
&\leq \exp\left( M(1+B)(1 - \int_0^t e^{s-t}\lambda_s ds) + M(1+\|x\|^2)(1 - \int_0^t e^{s-t}\lambda_s ds) \right)
\end{aligned}
$$

from which we obtain the result

$$
\begin{aligned}
D_{\mathrm{KL}}(\mu_t \| \pi) &\leq M(1+B)(1 - \int_0^t e^{s-t}\lambda_s ds) + M(1 + \int \mu_t(x)\|x\|^2 dx)(1 - \int_0^t e^{s-t}\lambda_s ds) \\
&\leq M(1+B)(1 - \int_0^t e^{s-t}\lambda_s ds) + M(1 + B\exp\left( M(1+B)(1-\int_0^t e^{s-t}\lambda_s ds) \right))(1 - \int_0^t e^{s-t}\lambda_s ds)
\end{aligned}
$$

using Proposition 8.

**Proposition 8.** *Assume that $\int_{\mathbb{R}^d} \|x\|^2 \mu_0(x) dx \leq B$, $\int_{\mathbb{R}^d} \|x\|^2 \pi(x) dx \leq B$ and equation 6, then*

$$
\int_{\mathbb{R}^d} \mu_t(x)\|x\|^2 dx \leq B\exp\left( M(1+B)(1 - \int_0^t e^{s-t}\lambda_s ds) \right).
$$

*Proof.* Using the expression for $\mu_t$ in equation 5 and Hölder inequality with $p = (1 - \int_0^t e^{s-t}\lambda_s ds)^{-1}$ and $q = (\int_0^t e^{s-t}\lambda_s ds)^{-1}$ gives

$$
\begin{aligned}
\int_{\mathbb{R}^d} \mu_t(x)\|x\|^2 dx &= \frac{1}{Z_t} \int_{\mathbb{R}^d} (\mu_0(x)\|x\|^2)^{1-\int_0^t e^{s-t}\lambda_s ds} (\pi(x)\|x\|^2)^{\int_0^t e^{s-t}\lambda_s ds} dx \\
&\leq \frac{1}{Z_t} \left( \int_{\mathbb{R}^d} \mu_0(x)\|x\|^2 dx \right)^{1-\int_0^t e^{s-t}\lambda_s ds} \left( \int_{\mathbb{R}^d} \pi(x)\|x\|^2 dx \right)^{\int_0^t e^{s-t}\lambda_s ds} \\
&\leq B\exp\left( M(1+B)(1 - \int_0^t e^{s-t}\lambda_s ds) \right).
\end{aligned}
$$

$\square$

### B.4 Proof of Proposition 3

For the geometric tempering family the KL divergence is given by

$$
D_{\mathrm{KL}}(\rho_\alpha \| \pi) = \int_{\mathbb{R}^d} \rho_\alpha(x) \log \frac{\rho_\alpha(x)}{\pi(x)} dx = (1-\alpha)\mathbb{E}_{\rho_\alpha}[s] - \log Z_\alpha
$$

where $s(x) := \log \frac{\mu_0(x)}{\pi(x)}$.

Differentiating $D_{\mathrm{KL}}$ w.r.t. $\alpha$ we find

$$
\frac{d}{d\alpha} D_{\mathrm{KL}}(\rho_\alpha \| \pi) = -(1-\alpha)\mathrm{Var}_{\rho_\alpha}[s] \leq 0.
$$

since

$$
\begin{aligned}
\frac{d}{d\alpha}(1-\alpha)\mathbb{E}_{\rho_\alpha}[s] &= -\mathbb{E}_{\rho_\alpha}[s] + (1-\alpha)\frac{d}{d\alpha}\mathbb{E}_{\rho_\alpha}[s] \\
&= -\mathbb{E}_{\rho_\alpha}[s] + (1-\alpha)\int_{\mathbb{R}^d} \frac{d}{d\alpha}\rho_\alpha(x)s(x)dx \\
&= -\mathbb{E}_{\rho_\alpha}[s] - (1-\alpha)\int_{\mathbb{R}^d} \rho_\alpha(x)\left[s(x) - \mathbb{E}_{\rho_\alpha}[s]\right]s(x)dx,
\end{aligned}
$$

where we used the fact that $\frac{d}{d\alpha}\rho_\alpha(x) = \rho_\alpha(x)\frac{d}{d\alpha}\log\rho_\alpha(x)$. Similarly, we obtain

$$\frac{d}{d\alpha}(-\log Z_\alpha) = \mathbb{E}_{\rho_\alpha}[s].$$

Integrating with respect to $\alpha$ from $\int_0^t e^{s-t}\lambda_s ds$ to $1 - e^{-t} \geq \int_0^t e^{s-t}\lambda_s ds$ we find

$$D_{\mathrm{KL}}(\mu_t^{\mathrm{FR}}||\pi) - D_{\mathrm{KL}}(\mu_t^{\mathrm{T\text{-}FR}}||\pi) = -\int_{\int_0^t e^{s-t}\lambda_s ds}^{1-e^{-t}} (1 - u)\,\mathrm{Var}_{\rho_u}\left(\log\frac{\mu_0}{\pi}\right)\,du \leq 0.$$

## C   Analytic solutions: Multivariate Gaussian case

This section focuses on comparing the evolution of the PDEs introduce in this section for the special case of transporting a Gaussian $\mu_0(x) = \mathcal{N}(x; m_0, \Sigma_0)$ to target Gaussian $\pi(x) = \mathcal{N}(x; m_\pi, \Sigma_\pi)$. In this case, we can obtain nice expressions for the moments along the flow described by each PDE.

In particular, observing that both the tempered W flow and the tempered FR flow preserve gaussianity, we have that $\mu_t(x) = \mathcal{N}(x; m_t, \Sigma_t)$ for each time $t \geq 0$.

We detail here the results needed to obtain the ODEs describing the tempered and standard PDEs for the special case of transporting a Gaussian $\mu_0(x) = \mathcal{N}(x; m_0, \Sigma_0)$ to target Gaussian $\pi(x) = \mathcal{N}(x; m_\pi, \Sigma_\pi)$.

**Wasserstein Flow (Langevin)**   The PDE in this case is the Fokker-Planck equation for the overdamped Langevin SDE

$$\begin{aligned}
dX_t &= -\nabla V_\pi(X_t)dt + \sqrt{2}dB_t \\
&= -\Sigma_\pi^{-1}(X_t - m_\pi)dt + \sqrt{2}dB_t.
\end{aligned}$$

Since the potential takes the form $V_\pi(x) = \frac{1}{2}(x - m_\pi)^\top \Sigma_\pi^{-1}(x - m_\pi)$, the moment equations are standard as this essentially boils down to an Ornstein-Uhlenbeck process. They are given by

$$\begin{aligned}
\dot{m}_t &= -\Sigma_\pi^{-1}(m_t - m_\pi) \\
\dot{\Sigma}_t &= -\Sigma_\pi^{-1}\Sigma_t - \Sigma_t\Sigma_\pi^{-1} + 2I
\end{aligned}$$

and the explicit solution can be found as

$$\begin{aligned}
m_t &= e^{-t\Sigma_\pi^{-1}}m_0 + (I - e^{-t\Sigma_\pi^{-1}})m_\pi \\
\Sigma_t &= e^{-t\Sigma_\pi^{-1}}\left(2\int_0^t e^{s\Sigma_\pi^{-1}}(e^{s\Sigma_\pi^{-1}})^\top ds + \Sigma_0\right)(e^{-t\Sigma_\pi^{-1}})^\top.
\end{aligned}$$

In the 1D case, we can easily evaluate the expression for $\Sigma_t$ as

$$\Sigma_t = e^{-2t\Sigma_\pi^{-1}}[\Sigma_0 + \Sigma_\pi(e^{2t\Sigma_\pi^{-1}} - 1)].$$

**Fisher–Rao flow**   In the Gaussian case, the ODEs for the mean and covariance respectively are given by

$$\begin{aligned}
\dot{m}_t &= -\Sigma_t\Sigma_\pi^{-1}(m_t - m_\pi) \\
\dot{\Sigma}_t &= -\Sigma_t\Sigma_\pi^{-1}\Sigma_t + \Sigma_t.
\end{aligned}$$

The explicit solutions in this case are known (see e.g. Chen et al. (2023, Lemma E.3)),

$$\begin{aligned}
m_t &= m_\pi + e^{-t}\left((1 - e^{-t})\Sigma_\pi^{-1} + e^{-t}\Sigma_0^{-1}\right)^{-1}\Sigma_0^{-1}(m_0 - m_\pi) \\
&= m_\pi + e^{-t}\Sigma_t\Sigma_0^{-1}(m_0 - m_\pi) \\
\Sigma_t^{-1} &= \Sigma_\pi^{-1} + e^{-t}(\Sigma_0^{-1} - \Sigma_\pi^{-1}).
\end{aligned}$$

**Tempered Wasserstein flow (Tempered Langevin; (Chehab et al., 2025))**  Recall the diffusion process characterising Tempered Langevin is given by

$$dX_t = -[(1-\lambda_t)\nabla_x V_0 + \lambda_t \nabla_x V_\pi]dt + \sqrt{2}dB_t$$

where $\lambda_t : \mathbb{R}_+ \to [0,1)$ is a non-decreasing function of $t$. The corresponding evolution PDE given by

$$\partial_t \mu_t(x) = \nabla \cdot \left( \mu_t(x)\nabla \log\left(\frac{\mu_t(x)}{\pi_t(x)}\right)\right) = \nabla \cdot \left( \mu_t(x)\nabla \log\left(\frac{\mu_t(x)}{\pi(x)}\right)^{\lambda_t}\right) + \nabla \cdot \left( \mu_t(x)\nabla \log\left(\frac{\mu_t(x)}{\mu_0(x)}\right)^{1-\lambda_t}\right)$$

where $\pi_t \propto \pi^{\lambda_t}\mu_0^{1-\lambda_t}$. The second equality shows that it can be seen as a sum of two Wasserstein parts with temperatures $\lambda_t$ and $1-\lambda_t$.

Note that $\pi_t$ is a product of 2 Gaussian densities $\mathcal{N}\left(m_\pi, \frac{\Sigma_\pi}{\lambda_t}\right)$ and $\mathcal{N}\left(m_0, \frac{\Sigma_0}{1-\lambda_t}\right)$ therefore $\pi_t \propto \mathcal{N}(a_t, P_t)$ where

$$a_t = \left( m_0\frac{\Sigma_\pi}{\lambda_t} + m_\pi\frac{\Sigma_0}{1-\lambda_t}\right)\left(\frac{\Sigma_0}{1-\lambda_t} + \frac{\Sigma_\pi}{\lambda_t}\right)^{-1}$$
$$P_t = \Sigma_\pi(\lambda_t\Sigma_0 + (1-\lambda_t)\Sigma_\pi)^{-1}\Sigma_0$$

and the intermediate densities are still Gaussian. Then we have the same structure for the moment ODEs as for infinite time Wasserstein gradient flow, except with $m_\pi, \Sigma_\pi$ replaced by $a_t, P_t$ respectively,

$$\dot{m}_t = -\left(\Sigma_\pi(\lambda_t\Sigma_0 + (1-\lambda_t)\Sigma_\pi)^{-1}\Sigma_0\right)^{-1}\left(m_t - \left(m_0\frac{\Sigma_\pi}{\lambda_t} + m_\pi\frac{\Sigma_0}{1-\lambda_t}\right)\left(\frac{\Sigma_0}{1-\lambda_t} + \frac{\Sigma_\pi}{\lambda_t}\right)^{-1}\right)$$

$$= -(\lambda_t\Sigma_\pi^{-1} + (1-\lambda_t)\Sigma_0^{-1})\left(m_t - ((1-\lambda_t)m_0\Sigma_\pi + \lambda_t m_\pi\Sigma_0)\left(\lambda_t\Sigma_0 + (1-\lambda_t)\Sigma_\pi\right)^{-1}\right)$$

$$\dot{\Sigma}_t = -\left(\Sigma_\pi(\lambda_t\Sigma_0 + (1-\lambda_t)\Sigma_\pi)^{-1}\Sigma_0\right)^{-1}\Sigma_t - \Sigma_t\left(\Sigma_\pi(\lambda_t\Sigma_0 + (1-\lambda_t)\Sigma_\pi)^{-1}\Sigma_0\right)^{-1} + 2I$$

$$= -(\lambda_t\Sigma_\pi^{-1} + (1-\lambda_t)\Sigma_0^{-1})\Sigma_t - \Sigma_t(\lambda_t\Sigma_\pi^{-1} + (1-\lambda_t)\Sigma_0^{-1}) + 2I.$$

In the 1D case, the explicit solution is given by

$$m_t = m_0 e^{-\int_0^t (1-\lambda_s)ds\Sigma_0^{-1} - \int_0^t \lambda_s ds\Sigma_\pi^{-1}}$$
$$+ e^{-\int_0^t (1-\lambda_s)ds\Sigma_0^{-1} - \int_0^t \lambda_s ds\Sigma_\pi^{-1}} \int_0^t [(1-\lambda_u)\Sigma_0^{-1}m_0 + \lambda_u\Sigma_\pi^{-1}m_\pi]e^{\int_0^u (1-\lambda_s)ds\Sigma_0^{-1} + \int_0^u \lambda_s ds\Sigma_\pi^{-1}} du$$

$$\Sigma_t = \Sigma_0 e^{-2\int_0^t (1-\lambda_s)ds\Sigma_0^{-1} - 2\int_0^t \lambda_s ds\Sigma_\pi^{-1}}$$
$$+ 2e^{-2\int_0^t (1-\lambda_s)ds\Sigma_0^{-1} - 2\int_0^t \lambda_s ds\Sigma_\pi^{-1}} \int_0^t e^{2\int_0^u (1-\lambda_s)ds\Sigma_0^{-1} + 2\int_0^u \lambda_s ds\Sigma_\pi^{-1}} du.$$

**Tempered Fisher–Rao flow**  Recalling that $\pi_t \propto \mathcal{N}(a_t, P_t)$ with $a_t, P_t$ as before and observing that tempered FR PDE can be written as

$$\partial_t \mu_t = \mu_t(x)\left(\log\left(\frac{\pi_t(x)}{\mu_t(x)}\right) - \mathbb{E}_{\mu_t}\left[\log\left(\frac{\pi_t}{\mu_t}\right)\right]\right)$$
$$= \mu_t(x)\left(\log\left(\frac{\pi(x)}{\mu_t(x)}\right)^{\lambda_t} - \mathbb{E}_{\mu_t}\left[\log\left(\frac{\pi}{\mu_t}\right)^{\lambda_t}\right]\right) + \mu_t(x)\left(\log\left(\frac{\mu_0(x)}{\mu_t(x)}\right)^{1-\lambda_t} - \mathbb{E}_{\mu_t}\left[\log\left(\frac{\mu_0}{\mu_t}\right)^{1-\lambda_t}\right]\right),$$

we see that the tempered FR PDE is the sum of two Fisher–Rao paths with temperatures $\lambda_t$ and $1-\lambda_t$.

The ODEs describing the evolution of the moments of $\mu_t$ are obtained from those for the standard FR flow replacing $m_\pi, \Sigma_\pi$ with $a_t, P_t$

$$\dot{m}_t = -\Sigma_t(\lambda_t\Sigma_\pi^{-1} + (1-\lambda_t)\Sigma_0^{-1})(m_t - m_\pi)$$
$$\dot{\Sigma}_t = -\Sigma_t(\lambda_t\Sigma_\pi^{-1} + (1-\lambda_t)\Sigma_0^{-1})\Sigma_t + \Sigma_t.$$

In the 1D case, the explicit solution is given by

$$m_t = m_\pi + \exp\left(-\int_0^t \Sigma_s\left(\lambda_s \Sigma_\pi^{-1} + (1-\lambda_s)\Sigma_0^{-1}\right)ds\right)(m_0 - m_\pi)$$

$$\Sigma_t = \frac{1}{e^{-t}\Sigma_0^{-1} + \int_0^t e^{-(t-s)}\left(\lambda_s \Sigma_\pi^{-1} + (1-\lambda_s)\Sigma_0^{-1}\right)ds}.$$

## D  Discrete time convergence

### D.1  Proof of Proposition 4

To deal with the tempered W flow update we decompose

$$D_{\mathrm{KL}}(\mu_n||\pi) = D_{\mathrm{KL}}(\mu_n||\pi) + D_{\mathrm{KL}}(\mu_n||\pi_n) - D_{\mathrm{KL}}(\mu_n||\pi_n) \tag{30}$$

$$= D_{\mathrm{KL}}(\mu_n||\pi_n) + \int_{\mathbb{R}^d} \mu_n(x)\log\frac{\pi_n(x)}{\pi(x)}dx.$$

We recall that $\mu_n$ is obtained as

$$\mu_n(x) = (4\pi\gamma_n)^{-d/2}\int_{\mathbb{R}^d}\exp\left(-\frac{1}{4\gamma_n}\|x - y - \gamma_n\nabla\log\pi_n(y)\|^2\right)\mu_{n-1}(y)dy$$

and using Lemma 4 with $\gamma_n \leq c_n/(4L_n^2)$ we obtain

$$D_{\mathrm{KL}}(\mu_n||\pi_n) \leq e^{-\gamma_n c_n}D_{\mathrm{KL}}(\mu_{n-1}||\pi_n) + 6\gamma_n^2 dL_n^2.$$

Thus we obtain

$$D_{\mathrm{KL}}(\mu_n||\pi) \leq e^{-\gamma_n c_n}D_{\mathrm{KL}}(\mu_{n-1}||\pi_n) + 6\gamma_n^2 dL_n^2 + \int_{\mathbb{R}^d}\mu_n(x)\log\frac{\pi_n(x)}{\pi(x)}dx$$

$$\leq e^{-\gamma_n c_n}D_{\mathrm{KL}}(\mu_{n-1}||\pi) + 6\gamma_n^2 dL_n^2 + \int_{\mathbb{R}^d}[\mu_{n-1}(x) - \mu_n(x)]\log\frac{\pi(x)}{\pi_n(x)}dx$$

$$= e^{-\gamma_n c_n}D_{\mathrm{KL}}(\mu_{n-1}||\pi) + 6\gamma_n^2 dL_n^2 + (1-\lambda_n)\int_{\mathbb{R}^d}[\mu_{n-1}(x) - \mu_n(x)]\log\frac{\pi(x)}{\mu_0(x)}dx,$$

where we used the fact that $D_{\mathrm{KL}}(\mu_{n-1}||\pi_n) = D_{\mathrm{KL}}(\mu_{n-1}||\pi) + \int_{\mathbb{R}^d}\mu_{n-1}\log\pi/\pi_n$ and $e^{-\gamma_n c_n} \leq 1$ and that $\pi_n = \mu_0^{1-\lambda_n}\pi^{\lambda_n}/\mathcal{Z}_n$.

Let us focus on

$$\int_{\mathbb{R}^d}[\mu_{n-1}(x) - \mu_n(x)]\log\frac{\pi(x)}{\mu_0(x)}dx = \int\mu_{n-1}(x)(V_0(x) - V_\pi(x))dx + \int_{\mathbb{R}^d}\mu_n(x)(V_\pi(x) - V_0(x))dx$$

$$\leq \int\mu_{n-1}(x)[V_0(x) - \inf_{x\in\mathbb{R}^d}V_\pi(x)]dx + \int_{\mathbb{R}^d}\mu_n(x)[V_\pi(x) - \inf_{x\in\mathbb{R}^d}V_0(x)]dx$$

$$\leq \int_{\mathbb{R}^d}\mu_{n-1}(x)[V_0(x) - \inf_{x\in\mathbb{R}^d}V_0(x)]dx + \int_{\mathbb{R}^d}\mu_n(x)[V_\pi(x) - \inf_{x\in\mathbb{R}^d}V_\pi(x)]dx$$

as $\inf_{x\in\mathbb{R}^d}V_\pi(x), \inf_{x\in\mathbb{R}^d}V_0(x)$ are constants. Using Lemma 2 under equation 27, we can bound

$$\int_{\mathbb{R}^d}[\mu_{n-1}(x) - \mu_n(x)]\log\frac{\pi(x)}{\mu_0(x)}dx \leq 2L_0\frac{b_0}{a_0} + 2L_\pi\frac{b_\pi}{a_\pi} + 2L_0\int_{\mathbb{R}^d}\mu_{n-1}(x)\|x\|^2 dx + 2L_\pi\int_{\mathbb{R}^d}\mu_n(x)\|x\|^2 dx$$

$$\leq 2L_0\frac{b_0}{a_0} + 2L_\pi\frac{b_\pi}{a_\pi} + 2(L_0 + L_\pi)\max\left(\mathbb{E}_{\mu_0}[\|x\|^2], \frac{2(3(b_0 + b_\pi)/2 + d)}{a_0 \wedge a_\pi \wedge 1}\right) := A'$$

using Lemma 3 to bound the second moment of $\mu_n$ for all $n \geq 0$ since $\gamma_n \leq \min(1, \frac{a_\pi \wedge a_0}{2(L_\pi + L_0)^2})$ for all $n \geq 0$.

Using induction we then obtain

$$D_{\mathrm{KL}}(\mu_n||\pi) \leq D_{\mathrm{KL}}(\mu_0||\pi)e^{-\sum_{k=1}^n \gamma_k c_k} + \sum_{k=0}^{n-1}(6\gamma_k^2 dL_k^2 + (1-\lambda_k)A')e^{-\sum_{i=1}^k \gamma_i c_i}.$$

## E  Auxiliary results for Section 3

### E.1  First variation of $\mathcal{F}$

We recall the definition of first variation of a functional $\mathcal{F} : \mathsf{E} \to \mathbb{R}^+$ on $\mathsf{E}$. We denote by $\mathsf{E}^*$ the dual of $\mathsf{E}$ and for any $z \in \mathsf{E}$ and $z^* \in \mathsf{E}^*$ we denote the dot product by $\langle z^*, z \rangle$.

**Definition 1** (Derivative). *If it exists, the derivative of $\mathcal{F}$ at $z_1$ is the function $\delta\mathcal{F}(z_1) : \mathsf{E}^* \to \mathbb{R}$ s.t. for any $z_2 \in \mathsf{E}$, with $\xi = z_2 - z_1$:*

$$\lim_{\epsilon \to 0} \frac{1}{\epsilon}(\mathcal{F}(z_1 + \epsilon\xi) - \mathcal{F}(z_1)) = \langle \delta\mathcal{F}(z_1), \xi \rangle,$$

*and is defined uniquely up to an additive constant.*

Using this definition for $\mathcal{F}(\mu, \lambda) = D_{\mathrm{KL}}(\mu||\pi_\lambda) + D_{\mathrm{KL}}(\pi_\lambda||\pi)$ we have for $\mu_i$, $i = 1, 2$, and $\xi_\mu = \mu_2 - \mu_1$,

$$\mathcal{F}(\mu_1 + \epsilon\xi_\mu, \lambda) - \mathcal{F}(\mu_1, \lambda) = \int_{\mathbb{R}^d} \log\left([\mu_1 + \epsilon\xi_\mu](x)\right)[\mu_1 + \epsilon\xi_\mu](x)dx - \int_{\mathbb{R}^d} \log\pi_\lambda(x)[\mu_1 + \epsilon\xi_\mu](x)dx +$$

$$+ \int_{\mathbb{R}^d} \log\pi_\lambda(x)\mu_1(x)dx - \int_{\mathbb{R}^d} \log\left(\mu_1(x)\right)\mu_1(x)dx$$

$$= -\epsilon\int_{\mathbb{R}^d} \log\pi_\lambda(x)\xi_\mu(x)dx + \epsilon\int_{\mathbb{R}^d} \log\left([\mu_1 + \epsilon\xi_\mu](x)\right)\xi_\mu(x)dx$$

$$+ \int_{\mathbb{R}^d} \log\left(1 + \epsilon\frac{\xi_\mu(x)}{\mu_1(x)}\right)\mu_1(x)dx.$$

We then have that

$$\lim_{\epsilon \to 0} \frac{1}{\epsilon}(\mathcal{F}(z_1 + \epsilon\xi) - \mathcal{F}(z_1)) = -\int_{\mathbb{R}^d} \log\pi_\lambda(x)\xi_\mu(x)dx + \int_{\mathbb{R}^d} \frac{\xi_\mu(x)}{\mu_1(x)}\mu_1(x)dx + \int_{\mathbb{R}^d} \log\left(\mu_1(x)\right)\xi_\mu(x)dx,$$

where the equality follows from the Taylor expansion of the logarithm as $\epsilon \to 0$. Therefore, we can write

$$\lim_{\epsilon \to 0} \frac{1}{\epsilon}(\mathcal{F}(z_1 + \epsilon\xi) - \mathcal{F}(z_1)) = \left\langle \log\frac{\mu_1(x)}{\pi_\lambda(x)} + 1, \xi_\mu \right\rangle.$$

The result follows since $\delta\mathcal{F}(\mu)/\delta\mu$ is defined up to additive constants.

### E.2  Proof of Proposition 6

Remind ourselves of the paired gradient flow system

$$\partial_t\mu_t(x) = -\mu_t(x)\left(\log\frac{\mu_t(x)}{\pi_{\lambda_t}(x)} - \mathbb{E}_{\mu_t}\left[\log\frac{\mu_t(x)}{\pi_{\lambda_t}(x)}\right]\right) \tag{31}$$

$$\dot{\lambda}_t = \int_{\mathbb{R}^d} \mu_t(x)\log\frac{\pi(x)}{\mu_0(x)}dx - \int_{\mathbb{R}^d} \pi_{\lambda_t}(x)\log\frac{\pi(x)}{\mu_0(x)}dx + (1-\lambda_t)\mathrm{Var}_{\pi_{\lambda_t}}\left(\log\frac{\mu_0}{\pi}\right). \tag{32}$$

For which we can instead write

$$\mu_t \propto \mu_0^{e^{-t}+e^{-t}\int_0^t e^s ds - \beta_t}\pi^{\beta_t} = \mu_0^{e^{-t}+e^{-t}(e^t-1)-\beta_t}\pi^{\beta_t} = \mu_0^{1-\beta_t}\pi^{\beta_t} \tag{33}$$

where

$$\beta_t = \int_0^t e^{s-t} \lambda_s ds$$

Noticing that equation 33 has the form of equation 9 with $\alpha = \beta_t$, so we now use the notation $\pi_{\beta_t}$ in place of $\mu_t$. The ODE for $\beta_t$ is given by

$$\dot{\beta}_t = -e^{-t} \int_0^t e^s \lambda_s ds + e^{-t} e^t \lambda_t = -\beta_t + \lambda_t.$$

Then using the shorthand notation $g_0 = \log \frac{\pi}{\mu_0}$, we have

$$\dot{\lambda}_t = \mathbb{E}_{\pi_{\beta_t}}[g_0] - \mathbb{E}_{\pi_{\lambda_t}}[g_0] + (1 - \lambda_t) \mathrm{Var}_{\pi_{\lambda_t}}(g_0).$$

Now for the case $g_0 = \log \frac{\pi}{\mu_0}$, it holds that

$$\frac{d}{d\alpha} \mathbb{E}_{\pi_\alpha}[g_0] = \mathrm{Var}_{\pi_\alpha}[g_0]$$

see Appendix B.4 for a derivation. So then the ODE simplifies to

$$\dot{\lambda}_t = \mathbb{E}_{\pi_{\beta_t}}[g_0] - \mathbb{E}_{\pi_{\lambda_t}}[g_0] + (1 - \lambda_t) \frac{d}{d\alpha} \mathbb{E}_{\pi_\alpha}[g_0]\Big|_{\alpha = \pi_{\lambda_t}}$$

So now we have a paired ODE system that can be understood purely in terms of $\beta_t, \lambda_t$ with $\mu_0, \pi$ as fixed constants. So we can instead check how $\mathcal{F}$ varies with $(\lambda, \beta)$, which allows us to avoid dealing with gradient flows in probability measure space. In order to prove asymptotic stability of $(\lambda, \beta) = (1, 1)$ by LaSalle invariance principle type arguments, we choose $\mathcal{F}(\lambda_t, \beta_t) = D_{\mathrm{KL}}(\pi_{\beta_t} || \pi_{\lambda_t}) + D_{\mathrm{KL}}(\pi_{\lambda_t} || \pi)$ as a Lyapunov functional and show

1. $\mathcal{F} \geq 0$ for all $\lambda, \beta \in (0, 1]$

2. $\mathcal{F} = 0$ iff $\lambda = 1, \beta = 1$

3. $\frac{d}{dt} \mathcal{F} \leq 0$ for all $\lambda, \beta \in (0, 1]$.

4. $\frac{d}{dt} \mathcal{F} = 0$ iff $\lambda = 1, \beta = 1$

The above conditions are sufficient since $\lambda, \beta \in [0, 1]$ and therefore vary in a compact interval. The first item trivially holds by non-negativity of the KL and second also holds trivially by the fact that $D_{\mathrm{KL}}(\pi_{\lambda_t} || \pi) = 0$ iff $\lambda_t = 1$ and therefore $\beta_t = 1$ also. For the third item, differentiating with respect to time yields

$$\frac{d}{dt} \mathcal{F}(\lambda_t, \beta_t) = \frac{d}{dt} D_{\mathrm{KL}}(\pi_{\beta_t} || \pi_{\lambda_t}) + D_{\mathrm{KL}}(\pi_{\lambda_t} || \pi)$$
$$= \int_{\mathbb{R}^d} \log \frac{\pi_{\beta_t}(x)}{\pi_{\lambda_t}(x)} \partial_t \pi_{\beta_t}(x) dx - \int_{\mathbb{R}^d} \pi_{\beta_t}(x) \partial_t \log \pi_{\lambda_t}(x) dx + \frac{d}{d\lambda_t} D_{\mathrm{KL}}(\pi_{\lambda_t} || \pi) \dot{\lambda}_t.$$

Using the derivative of KL w.r.t. $\lambda$ derived in Appendix B.4, the fact that

$$\partial_\lambda \log \pi_\lambda(x) = \log \frac{\pi(x)}{\mu_0(x)} - \int_{\mathbb{R}^d} \pi_\lambda(x) \log \frac{\pi(x)}{\mu_0(x)} dx,$$

we obtain

$$\frac{d}{dt} \mathcal{F}(\lambda_t, \beta_t) = \int_{\mathbb{R}^d} \log \frac{\pi_{\beta_t}(x)}{\pi_{\lambda_t}(x)} \partial_t \pi_{\beta_t}(x) dx - \int_{\mathbb{R}^d} \pi_{\beta_t}(x) \partial_t \log \pi_{\lambda_t}(x) dx + \frac{d}{d\lambda_t} D_{\mathrm{KL}}(\pi_{\lambda_t} || \pi) \dot{\lambda}_t$$
$$= -\mathrm{Var}_{\pi_{\beta_t}} \left( \log \frac{\pi_{\beta_t}}{\pi_{\lambda_t}} \right) - (\dot{\lambda}_t)^2 < 0, \tag{34}$$

showing that $\frac{d}{dt}\mathcal{F}(\lambda_t, \beta_t) \leq 0$ for any $\lambda_t, \beta_t \in [0,1]$. Furthermore, the rate of decay of $\mathcal{F}$ is driven both by the tempered FR PDE and the optimally chosen $\lambda_t$. Finally, we check if $(\lambda, \beta) = (1,1)$ is a unique fixed point of equation 16 and equation 17, that is, to find $\lambda^*, \beta^*$ such that $\frac{d}{dt}\mathcal{F}(\lambda^*, \beta^*) = 0$. For this, we require individually,

$$\text{Var}_{\pi_{\beta_t}}\left(\log \frac{\pi_{\beta_t}}{\pi_{\lambda_t}}\right) = 0; \quad \dot{\lambda}_t = 0$$

based on equation 34. Starting with the first condition, this can happen only when $\log \frac{\mu_t}{\pi_{\lambda_t}} = c$ for all $x \in \mathbb{R}^d$, which can only happen when $\pi_{\beta_t} = \pi_{\lambda_t}$, meaning $\beta_t = \lambda_t$. Then substitute this into the ODE for $\lambda_t$ to obtain

$$\dot{\lambda}_t = \int_{\mathbb{R}^d} \pi_{\lambda_t}(x) \log \frac{\pi(x)}{\mu_0(x)} dx - \int_{\mathbb{R}^d} \pi_{\lambda_t}(x) \log \frac{\pi(x)}{\mu_0(x)} dx + (1-\lambda_t)\text{Var}_{\pi_{\lambda_t}}\left(\log \frac{\mu_0}{\pi}\right)$$
$$= (1-\lambda_t)\text{Var}_{\pi_{\lambda_t}}\left(\log \frac{\mu_0}{\pi}\right).$$

Clearly $\text{Var}_{\pi_{\lambda_t}}\left(\log \frac{\mu_0}{\pi}\right)$ cannot be zero unless $\pi_{\lambda_t}$ is the zero function. Therefore, $\lambda_t = 1$, which also then means that $\beta_t = 1$.

### E.3 Proof of Proposition 7

We proceed by the same reasoning as in Appendix E.2 with one slight difference, as we do not have access to a closed form solution to the W flow. We present a similar La Salle invariance principle type argument, with the main difference being that we must establish that $\mu_t$ trajectories from solving equation 19 are relatively compact in $\mathcal{P}_2^{ac}(\mathbb{R}^d)$.

Firstly, we trivially have that $\mathcal{F} \geq 0$ for all $\mu \in \mathcal{P}_2^{ac}(\mathbb{R}^d), \lambda \in (0,1]$ and also $\mathcal{F} = 0$ iff $\lambda_t = 1, \mu_t = \pi$. It straightforwardly follows from equation 20 that $\frac{d\mathcal{F}}{dt} \leq 0$ for all $\mu \in \mathcal{P}_2^{ac}(\mathbb{R}^d), \lambda \in (0,1]$. We also have from the same reasoning as in Appendix E.2 that $\frac{d\mathcal{F}}{dt} = 0$ iff $(\lambda_t, \mu_t) = (1, \pi)$. Finally, in order to obtain asymptotic stability, we require that trajectories $\{\mu_t, \lambda_t\}_{t \geq 0}$ are relatively compact. This trivially holds for $\lambda_t$ as it is confined to $(0,1]$. Now we have that $\{\mu_t\}_{t>0} \in \mathcal{P}_2^{ac}(\mathbb{R}^d)$ due to Lemma 1 in Appendix A and under Assumption 1 and $\mu_0 \in \mathcal{P}_2^{ac}(\mathbb{R}^d)$. A direct application of Proposition 7.1.5 in Ambrosio et al. (2008) yields the required relative compactness of trajectories of $\mu_t$.

