# OpenReview forum: "Properties and limitations of geometric tempering for gradient flow dynamics"
_TMLR — Accepted by TMLR_

### Review · Reviewer_M4RS · 2026-03-01

**Summary Of Contributions:**

The goal is to sample from a target density that is known up to a normalizing constant.

A popular sampling algorithm is the gradient flow of the KL divergence in the space of probability measures endowed with the Wasserstein metric, more commonly known as *Langevin dynamics*. Instead of applying this algorithm to sample directly from a target distribution, it is popular to apply it to sample from a sequence of distributions that terminate at the target distribution. This sequence is obtained as geometric means between a reference distribution and the target distribution. The final algorithm is called *Tempered Langevin dynamics*.

This paper extends previous analyses of the Tempered Langevin dynamics [1], by:
- **changing the metric** in the space of probability measures, from Wasserstein to Fisher-Rao (section 2.2) or Wasserstein-Fisher-Rao (section 2.3)
- **co-optimizing the tempering schedule**, which is the speed at which the sequence of distributions is traversed, as part of the gradient flow in the Wasserstein metric (section 3.1) or Fisher-Rao metric (section 3.2)

The paper extends the conclusion from [1] to these new setups: tempering does not accelerate convergence.

[1] Chehab et al. Provable Convergence and Limitations of Geometric Tempering for Langevin Dynamics. ICLR, 2025.

**Audience:**

Yes

**Audience Explanation:**

I believe some individuals in the TMLR audience would be interested in the toolbox used in this paper: using gradient flows and functional inequalities to obtain convergence rates of modern sampling algorithms that involve tempering.

**Clearer motivations**. Having said the above, I believe the motivation of this paper could be made clearer. Previous works [1, 2] have already stated a negative result: the geometric-mean interpolation is not "friendly" for sampling algorithms that locally transport mass. The authors here extend this negative result to new setups. One of them is a Fisher-Rao gradient flow whose implementation consists of a birth-death process: if a sample finds a new mode, the birth-death process can teleport more samples to that new mode, but finding a new mode remains hard. For this reason, it is not surprising that the negative result (slow convergence) holds in this new setup, which the authors quantitatively show in this paper. It would be helpful to add a few sentences to guide the reader in understanding why extending the analysis of [2] to these new setups (Fisher-Rao, Wasserstein-Fisher-Rao) is interesting or relevant, beyond being a mathematical exercise. Are the conclusions surprising or not? Is the main contribution the quantitative convergence rates?

[1] Mate et al. Learning Interpolations between Boltzmann Densities. TMLR, 2024.

[2] Chehab et al. Provable Convergence and Limitations of Geometric Tempering for Langevin Dynamics. ICLR 2025.

**Claims And Evidence:**

Yes

**Claims Explanation:**

The exposition seems mathematically accurate, but the claim that "tempering does not improve convergence" remains a bit unclear in light of how the results are stated.

**Q1. How much slower is the tempered version of Eq 7?**

Convergence is measured using an upper-bound as a function of time $t$. Consider Eq 7 for instance and suppose that the upper-bound is tight. The authors claim that the upper bound with tempering (Eq 7) decreases more slowly than the upper bound without tempering. It is unclear by what order these upper bounds differ: by an additive constant? a multiplicative constant? do both reach constant error in a time that is exponential in the distance between the target modes $m$, but one time is $\exp(4 m)$ and the other is $\exp(8 m^2)$ for example?
It would be useful to have a quantitative notion of "how much slower" the tempered version is.

**Q2. How much slower is the tempered version of section 3.1?**

The convergence of the tempered algorithm in section 3.1, where the tempering schedule is co-optimized, is given in Eq 16. I understand that upper-bound is comparable to the un-tempered Langevin but this is not clearly stated for the reader. So from the text only, it is unclear how Eq 16 compares with the tempered algorithm in section 2.1, where the tempering schedule is fixed. Or how it compares with the untempered Wasserstein Gradient Flow algorithm. This makes it hard to relate this result to the claim that "tempering does not improve convergence".

Moreover, the claim that "the Wasserstein gradient flow convergences exponentially fast", stated just before Eq 16, is a bit confusing. While it is mathematically true, the convergence rate depends on log-Sobolev constant of the target distribution $c_\pi$, which worsens exponentially in the between-mode distance, like the standard Langevin algorithm.

The same discussion applies to Proposition 5. How does the convergence upper-bound compare to the one in Proposition 2? and to the un-tempered Fisher-Rao gradient flow?

**Q3. Is $c_\pi$ the log-concavity constant or the log-Sobolev constant (more general) of the target distribution?**

This changes how widely applicable the results are. In Proposition 4, $c_\pi$ is a log-concavity constant and section 4.2 begins by saying that Proposition 1 applies to "strongly log-concave targets". However, in Assumption 1 $c_\pi$ seems to be a more general log-Sobolev constant. This interchangeability in vocabulary appears in other places in the paper.

**Q4. How easy or hard is it to implement in practice the schedule optimization in section 3?**

Section 3 is dedicated to co-optimizing the tempering schedule, that is, the speed of traversal of the sequence of distributions ends with the target distribution. There is an entire literature on optimizing the tempering schedule, in diffusion models and in annealed importance sampling, for instance. It would be useful for the authors to contextualize their results. How do the schedule updates, for example $\dot{\lambda}$ in Eq 15, compare to known ones? And especially, how easily implementable are they: do they lead to a new algorithm where the tempering schedule $\dot{\lambda}$ can be adapted "on the fly"?

**Requested Changes:**

Could the authors address my questions (Q1, Q2, Q3, Q4) and paragraph on "clearer motivations"; both are stated above. These would simply strengthen the work in my view.

---

> ### Author Response · Authors · 2026-03-23
> **Rebuttal by authors**
>
> We thank the reviewer for their valuable comments and questions. Below we propose our modifications, we appreciate any feedback while we continue to revise the manuscript.
>
> ## Reply to questions
>
> - **Q1** As both the tempered and standard FR flow have exact solutions which belong to the geometric tempering family, it is possible to exactly quantify the difference in KL at time $t$
>     \begin{align*}
>      \KL(\mu_t^{\textrm{FR}} || \pi)-\KL(\mu_t^{\textrm{T-FR}} || \pi)= -\int_{\int_0^te^{s-t} \lambda_s ds}^{1-e^{-t}} (1-u)\, \mathrm{Var}_{\rho_u}\left(\log\frac{\mu_0}{\pi}\right)\,du \le 0,
>     \end{align*}
>     and to show that the tempered FR flow \textbf{never} improves the KL decay w.r.t. the untempered FR flow. To the best of our knowledge, this is the first result giving an explicit comparison result for tempered and untempered dynamics.
> - **Q2**  Thank you for raising this point. We realised that the functional $\cF(\mu, \lambda):= \KL(\mu|| \pi_\lambda)$ considered before was a poor choice as it admits more than one global minimiser.  We will update section 3 to consider the functional $\cF(\mu, \lambda):= \KL(\mu|| \pi_\lambda)+\KL(\pi_\lambda||\pi)$.  We demonstrate that this functional has a unique minimiser at  $(\pi, 1)$ and that this point is asymptotically stable, so that $(\mu_t, \lambda_t) \rightarrow (\pi, 1)$ as $t \rightarrow \infty$ via LaSalle's principle.
>     In the case of the FR flow, our result in the reply to Q1 shows that the convergence of any tempered FR flow will always be slower. We check this is the case also for the ODE based $\lambda_t$ with an example which confirms our previous result.
>     For the W flow, we compare several adaptive schedules empirically and show that the one derived from the gradient flow perspective is typically slower then other alternatives in the literature as we will show in Section 4.4
> - **Q3** We apologise for the confusion, $c_\pi$ is the log-concavity constant. This stronger assumption is necessary to control the Laplacian in the proof of Proposition 1. We will clarify this just below assumption 1.
> - **Q4**  We will expand on this point by comparing with the ESS-based adaptive schedule choice popular in the SMC/AIS literature and the adaptive schedule of [1]. While a general result establishing the ``best'' schedule is out of the scope of this work, we observe empirically that the adaptive schedule in eq (15) tends to be slower than other alternatives in the literature. We believe this is due to the fact that to identify a gradient flow structure for the tempered PDEs we consider the functional $\cF(\mu, \lambda):= \KL(\mu|| \pi_\lambda)+\KL(\pi_\lambda||\pi)$, while commonly used approaches can be linked to the time derivative of $\KL(\mu_t||\pi)$ when $\mu_t$ follows the FR gradient flow.
>     In terms of computational tractability, all schedules provide some challenges. The ESS based one and the one in [3] are solutions of $$\frac{d}{dt} \KL(\pi_{\lambda_t}||\pi) = -(1-\lambda_t) Var \left[ \log  \frac{ \mu_0}{\pi}\right] \dot{\lambda}_t=\beta$$
> i.e. constant KL decay. However, as it requires knowledge of $\pi_{\lambda_t}\propto \pi^{\lambda_t}\mu_0^{1-\lambda_t}$ which is known up to a normalising constant, computation of the variance is intractable.
> The classical route is to assume that the samples that track the evolution of $\pi_{\lambda_t}$ provide a good approximation and thus use $\mu_t$ as an approximation of $\pi_{\lambda_t}$.
> In this case the schedule in Eq (15) becomes
> \begin{align}
>     \dot{\lambda}_t&= (1-\lambda_t) Var\left(\log\frac{\mu_0}{\pi}\right)
> \end{align}
> and does not correspond to constant KL decay. We will provide some numerical experiments investigating the differences between these schedules.
>
> [1] Shirin Goshtasbpour, Victor Cohen, and Fernando Perez-Cruz. Adaptive annealed importance sampling
> with constant rate progress. In Andreas Krause, Emma Brunskill, Kyunghyun Cho, Barbara Engelhardt,
> Sivan Sabato, and Jonathan Scarlett (eds.), Proceedings of the 40th International Conference on Machine
> Learning, volume 202 of Proceedings of Machine Learning Research, pp. 11642–11658. PMLR, 23–29 Jul
> 2023.

---

> ### Author Response · Authors · 2026-03-23
> **Clearer motivations**
>
> We agree that the motivation and main contributions could be made clearer. We believe that assessing tempering schedules for use in methods based on reweighting (as opposed to locally transporting mass as in [1], [2]) is important as there may be a different sequence of distributions that are well suited to reweighing rather than transportation methods. This is in particular based on knowledge from the SMC literature, in which adaptive tempering schedules are known to work well.
>
> Fisher-Rao gradient flows are already a kind of geometric tempering, so the question is to what extent does adding further tempering schedules improve or degrade convergence.  We believe the answer to this question is not immediately obvious based on findings for W flows which have very different mechanisms (transport vs reweighting).  We will improve on our previous result to now include an exact comparison of tempered and untempered FR flows (in a new proposition 3).  This new result shows that any choice of further tempering degrades performance (in terms of convergence time) compared to a pure FR flow.  We believe this is true for different reasons than in the W flow case: the slow down in the W flow is mostly due to the presence of a bias due to the moving targets $\pi_t$ approaching $\pi$ only asymptotically, in the case of the FR flow the slow down is not due to a bias term but rather to a slower exponent ($\int_0^t\lambda_s e^{t-s}ds$ instead of $1-e^{-t}$) of $\pi$ in the exact solution.
>
> We will also add a new section that explores jointly optimising tempering schedules via a gradient flow and comparing this to existing tempering schedules in the literature. Somewhat surprisingly, the solution of the gradient flow approach is slower than other approaches commonly used in the literature. We believe this is due to the fact that to identify a gradient flow structure for the tempered PDEs we consider the functional $\cF(\mu, \lambda):= \KL(\mu|| \pi_\lambda)+\KL(\pi_\lambda||\pi)$, while commonly used approaches can be linked to the time derivative of $\KL(\mu_t||\pi)$ when $\mu_t$ follows the FR gradient flow. In summary, our main contributions are:
>
> - Tempering dynamics never speed up convergence in the FR flow. This applies both to continuous time and to discrete time dynamics.
> - We will add a clearer comparison with Chehab et. al explaining how our results for the tempered W case differ from theirs. In particular, our Proposition 1 and 4 clearly shows that the upper bound on KL decay is given by a first term corresponding to the rate of the classical W PDE, to which a bias term is added due to the fact that $\pi_t$ approaches $\pi$ only as $t\to\infty$. This decomposition is different from the one in Chehab et al., Theorem 1 and 3.
> - We investigate a gradient flow based approach to identify optimal tempering schedules and compare them to standard tempering approaches.  Somewhat surprisingly, the solution of the gradient flow approach is slower than other approaches commonly used in the literature. We believe this is due to the fact that to identify a gradient flow structure for the tempered PDEs we consider the functional $\cF(\mu, \lambda):= \KL(\mu|| \pi_\lambda)+\KL(\pi_\lambda||\pi)$, while commonly used approaches can be linked to the time derivative of $\KL(\mu_t||\pi)$ when $\mu_t$ follows the FR gradient flow. We will add this comparison to the paper.

---

### Review · Reviewer_SRCz · 2026-03-09

**Summary Of Contributions:**

**Contributions**:

This paper follows the analysis in Chehab et al. (2025) to study the effect of geometric tempering on the convergence of gradient flow dynamics, including W, FR, and WFR gradient flows. The authors show that geometric tempering does not help to speed up convergence, both analytically and empirically.

**Strengths**:

- The paper studies an important question about the effect of geometric tempering, following the analysis in Chehab et al. (2025). It helps understand the limitations of geometric tempering, a widely used technique in sampling.
- The paper is well-written, and the results are clearly presented. There's no much difficulty in following the main arguments in the paper, except for section 3 which I'm not very familiar with.
- Numerical results, either by direct calculation following the PDE or by simulation of the dynamics, are well-designed and support the theoretical findings.

**Weaknesses**:

- The main technical tools used in the paper are standard calculations of the time derivative of the KL divergence and the use of LSI for bounding the convergence rate, and the discrete-time analysis is also classical. Thus, the paper does not contain too much technical novelty. Also, the limitations of geometric tempering are well-known (e.g., in Chehab et al. (2025) and [Guo et al. (2025, proposition 1)](https://arxiv.org/abs/2502.04575)), so the main contribution of the paper is to provide a further understanding beyond the W gradient flow (Langevin dynamics), which is a bit incremental.
- It would be more convincing if the paper can provide some lower bounds on the convergence rate of the tempered dynamics in some specific non-log-concave settings, like the analysis in Chehab et al. (2025, theorem 4). The current analysis only shows that the upper bound is not improved, and numerical results show that the convergence can be much slower.
- The conclusion section of the paper is not very compact and lacks a concise summary of the main findings.

**Audience:**

Yes

**Audience Explanation:**

The paper is of interest to the sampling community.

**Claims And Evidence:**

Yes

**Claims Explanation:**

The paper is mostly clear and convincing. The following are some specific comments:

- In figure 1: Should $C_t$ be $\Sigma_t$ instead? Is KL divergence calculated from the distribution of $X_t$ to $\pi_t$?
- In the above figure, the authors provided two cases with different convergence properties. I'll be happy to see some more systematic study of when W gradient flow is faster than the FR one and when it's not.
- The results on discrete-time FR gradient flow should be summarized into a proposition. Also need to clarify that this is simply the time-discretization of the PDE, not an algorithm that can be simulated.
- In section 3, is $\partial_\lambda \mathcal{F}(\mu,\lambda)$ always positive? Can we guarantee that when $t\to\infty$, $\lambda_t$ will converge to 1?
- In section 4.1, the simulation of WFR gradient flow is by a W flow + a resampling step. I'm not quite sure if this is the standard way to discretize the WFR flow, but it is possible to do in the reversed direction?
- Could the authors provide some evidence of the ESS on both original and tempered dynamics? Also in figure 3 what's the number of particles used in the simulation?

**Requested Changes:**

See "Are the claims made in the submission supported by accurate, convincing and clear evidence?" part.

---

> ### Author Response · Authors · 2026-03-18
> **Rebuttal by Authors**
>
> We thank the reviewer for their positive comments and relevant questions that will improve the paper. Below we propose our modifications, we appreciate any feedback while we continue to revise the manuscript.
>
> ## Weaknesses
>
> - We understand the referee's comment. However, given the increasing interest in FR and WFR flows [1, 2, 3, 4, 5 and many more] we believe that this paper contributes to understanding which strategies can or cannot be used to improve the performance of these sampling algorithms.
>     In addition, our results for the tempered W case differ from those in Chehab et al. Our Proposition 1 and 4 clearly shows that the upper bound on KL decay is given by a first term corresponding to the rate of the classical W PDE, to which a bias term is added due to the fact that $\pi_t$ approaches $\pi$ only as $t\to\infty$. This decomposition is different from the one in Chehab et al., Theorem 1 and 3.
> - We agree that focusing on the upper bound is not sufficient.  We note in the case of tempered FR dynamics, we can obtain an exact comparison result, of the form
> \begin{align*}
>      \KL(\mu_t^{\textrm{FR}} || \pi)-\KL(\mu_t^{\textrm{T-FR}} || \pi)= -\int_{\int_0^te^{s-t} \lambda_s ds}^{1-e^{-t}} (1-u)\, \mathrm{Var}_{\rho_u}\left(\log\frac{\mu_0}{\pi}\right)\,du \le 0,
> \end{align*}
> from which we conclude that tempered FR flows are always slower than FR flows.  We will add this as Proposition 3 to the revised manuscript.
> - We will improve the conclusions summarising the main takeaways and the motivation of this work, which are:
>     + Tempering dynamics never speed up convergence in the FR flow. This applies both to continuous time and to discrete time dynamics.
>     + We will add a clearer comparison with Chehab et. al explaining how our results for the tempered W case differ from theirs. In particular, our Proposition 1 and 4 clearly shows that the upper bound on KL decay is given by a first term corresponding to the rate of the classical W PDE, to which a bias term is added due to the fact that $\pi_t$ approaches $\pi$ only as $t\to\infty$. This decomposition is different from the one in Chehab et al., Theorem 1 and 3.
>     + We investigate a gradient flow based approach to identifying optimal tempering schedules and compare them to standard tempering approaches.  Somewhat surprisingly, the solution of the gradient flow approach is slower than other approaches commonly used in the literature. We believe this is due to the fact that to identify a gradient flow structure for the tempered PDEs we consider the functional $\cF(\mu, \lambda):= \KL(\mu|| \pi_\lambda)+\KL(\pi_\lambda||\pi)$, while commonly used approaches can be linked to the time derivative of $\KL(\mu_t||\pi)$ when $\mu_t$ follows the FR gradient flow. We will add this comparison to the paper.

---

> ### Author Response · Authors · 2026-03-18
> **Reply to requested changes**
>
> ## Reply to specific comments
>
> - Yes, $C_t$ should be replaced by $\Sigma_t$, thank you for bringing this to our attention. The KL divergence displayed is $\KL(\mu_t|\pi)$ where $\mu_t$ is the exact solution of the tempered flows (1), (4) and (8). We will make this clear in the revision.
> - We will include discussion about known theoretical results on cases when W flows have improved upper bound convergence rates in terms of $\KL(\mu_t||\pi)$ than FR (and vice versa).  Specifically, FR flows are guaranteed to converge exponentially fast at rate $e^{-t}$ under bounded second moment assumptions [5, Theorem 4.1].  The W flow converges exponentially fast at rate $e^{-t C_{LSI}^{-1}}$ for targets which satisfy a log-Sobolev inequality with constant $C_{LSI}$.  So when $C_{LSI} > 1$, the rate  of the upper bound is improved for W flows compared to FR flows (and vice versa when $C_{LSI} < 1$).
>     We feel that an empirical verification of this phenomenon is out of the scope of our work. Additionally, [1] compares numerical approximations of the W, FR and WFR flows on a wide range of targets.
> - We will add a proposition which summarises the results for the discrete time tempered FR flow. We also make clear that this time discretisation does not correspond (yet) to an algorithm.
> - Thank you for raising this point. We realised that the functional $\cF(\mu, \lambda):= \KL(\mu|| \pi_\lambda)$ considered before was a poor choice as it admits more than one global minimiser.  We will update section 3 to consider the functional $\cF(\mu, \lambda):= \KL(\mu|| \pi_\lambda)+\KL(\pi_\lambda||\pi)$.  We demonstrate that this functional has a unique minimiser at  $(\pi, 1)$ and that this point is asymptotically stable, so that $(\mu_t, \lambda_t) \rightarrow (\pi, 1)$ as $t \rightarrow \infty$ via LaSalle's principle. Additionally, we provide numerical evidence in the Gaussian case.
> - Yes it is indeed possible to swap the ordering of the W flow + resampling (FR flow) steps.  Algorithm 1 is an approximation of a splitting scheme for the WFR PDE in which the W and FR flow are solved sequentially.  The splitting scheme obtained by swapping the order of W and FR flow is also a valid approximation of the WFR PDE. We choose the ordering in Algorithm 1 as this is the one proposed in [1].
> - The number of particles in Figure 3 is $N=400$. Regarding the ESS, we would like to clarify whether you would like to see a comparison of ESS for tempering SMC and tempered SMC-WFR at the final time step? Or how the ESS evolves along the dynamics?
>
>
> [1] Crucinio \& Pathiraja, Sequential Monte Carlo approximations of Wasserstein–Fisher–Rao gradient flows (2025)
>
> [2] Carrillo et al., Fisher-Rao Gradient Flow: Geodesic Convexity and Functional Inequalities (2024)
>
> [3] Maurais \& Marzouk, Sampling in Unit Time with Kernel Fisher-Rao Flow (2024)
>
> [4] N\"usken, Stein transport for Bayesian inference (2024)
>
> [5] Yifan Chen, Daniel Zhengyu Huang, Jiaoyang Huang, Sebastian Reich, and Andrew M Stuart. Sampling via gradient flows in the space of probability measures. (2023)

---

### Review · Reviewer_bGBj · 2026-03-12

**Summary Of Contributions:**

The paper studies the tempered versions of Wasserstein, Fisher-Rao, and Wasserstein-Fisher-Rao dynamics. Upper bounds for the convergence of the continuous dynamics in KL divergence are provided for the first two dynamics, while the result for the third is a simple combination of the previous two. All the results suggest that tempering does not improve the convergence speed. In Section 2.4, an explicit calculation of the dynamics demonstrates that, for Gaussian targets, tempering slows down the dynamics.

The authors also provide a theoretical analysis of the tempered Unadjusted Langevin Algorithm (ULA), which includes an additional term compared to the standard ULA.

Since tempering introduces an additional mixture parameter, $\lambda$, the tempered dynamics are no longer gradient flows of the KL divergence functional. Instead, they become gradient flows in the extended space $\mathcal{P}(\mathbb{R}^d) \times [0,1]$. The authors show that, under the paper’s assumptions, results from prior works can be applied to provide descent for the functional $\mathcal{F}(\mu, \lambda) := D_{KL}(\mu \parallel \pi_\lambda)$.

The paper concludes with some experimental results.

**Additional Comments:**

The proofs are difficult to verify due to the omission of many details. While the results are likely correct, they do not fully justify the claims made. The writing of the paper is poor.

**Audience:**

Yes

**Audience Explanation:**

- The audience may be interested in the findings, provided they are properly contextualized within the existing literature and supported by rigorous proofs (see the "Requested Changes" section).

**Broader Impact Concerns:**

The submission is theoretical, there are no ethical concerns.

**Claims And Evidence:**

No

**Claims Explanation:**

**The method of the paper is counterintuitive.**
 The results presented in the paper are primarily negative. The authors argue that tempering does not accelerate continuous dynamics such as Wasserstein (W), Fisher-Rao (FR), or Wasserstein-Fisher-Rao (WFR) flows. However, all the results provided are upper bounds. To rigorously support such a claim, the authors should also provide lower bounds for the tempered dynamics.

In Section 2.4, the authors present an example involving Gaussians to demonstrate that tempered dynamics are slow. However, this example alone is insufficient as justification. The authors state:
> "Our results, as well as those in Chehab et al. (2025), suggest that tempered dynamics do not improve upon the gradient flow dynamics discussed in the previous section. However, our convergence results only obtain upper bounds for the decay of $D_{KL}$​, which are not necessarily tight. To confirm that tempered PDEs do not improve over the standard ones, we consider the special case of transporting Gaussians..."

While explicit computation shows that tempering does not improve upon standard gradient flows in this case, this does not confirm that the theoretical upper bound is tight. There may exist other classes of distributions where the W, FR, or WFR gradient flow is equivalent to or slower than its tempered alternative.


**Some of the mathematical claims lack sufficient detail:**

- The equation immediately following Equation (4) is not a fact but a consequence of (4). This should be clearly explained.
- The derivation of the analytic solution for the FR PDE is not detailed. Additionally, some integrals are missing their upper and lower bounds.
- On page 4, the statement "showing that the W and FR geometries are indeed orthogonal..." is unclear. What does "orthogonal" mean in this context?
- The paper frequently assumes familiarity with Chehab et al. (2025). For example, `Equation (27) is presented as an application of Step 2 of the proof of Theorem 3 in Chehab et al. (2025),` but this is not adequately explained in the current submission. Or, `This result is similar to Chehab et al. (2025, Theorem 3) and carries the same message.`Which message?
- The organization of the paper could be improved. For instance, Proposition 1 is stated without mentioning that its proof is in the appendix. Conversely, the proof in the appendix does not reference the statement it is proving. It would be clearer to cross-reference statements and proofs.
- Given that the proofs are not overly long or technical, mathematical details should not be omitted. For example, the computation of the decay of $D_{KL}$​ in Section A.1 should be thoroughly explained.
- The proofs in the appendix often directly apply lemmas from other papers (e.g., Chehab et al. 2025; Vempala and Wibisono, 2019) without properly stating them, making the proofs difficult to follow and verify.
-  On page 17, the following bound seems incorrect. There may be a typo,  $\inf_{x \in \mathbb{R}^d} V_0(x)$ and $\inf_{x \in \mathbb{R}^d} V_{\pi}(x)$ shall be interchanged
 $$\\begin{align*}
\\ldots &= \\int \\mu_{n-1}(x)\\big(V_0(x) - V_\\pi(x)\\big)dx + \\int \\mu_n(x)\\big(V_\\pi(x) - V_0(x)\\big)dx \\\\
&\\leq \\int \\mu_{n-1}(x)\\Big[V_0(x) - \\inf_{x \\in \\mathbb{R}^d} V_0(x)\\Big]dx + \\int \\mu_n(x)\\Big[V_\\pi(x) - \\inf_{x \\in \\mathbb{R}^d} V_\\pi(x)\\Big]dx.
\\end{align*}$$

**References**
- Omar Chehab, Anna Korba, Austin J Stromme, and Adrien Vacher. Provable convergence and limitations of geometric tempering for langevin dynamics. In The Thirteenth International Conference on Learning Representations, 2025. URL https://openreview.net/forum?id=DZcmz9wU0i.
- Santosh Vempala and Andre Wibisono. Rapid convergence of the unadjusted langevin algorithm: Isoperimetry suffices. In H. Wallach, H. Larochelle, A. Beygelzimer, F. d'Alché-Buc, E. Fox, and R. Garnett (eds.),Advances in Neural Information Processing Systems, volume 32. Curran Associates, Inc., 2019.

**Requested Changes:**

- **Improve rigor:** See the above sections for details.
- **Clarify the motivation:** The paper should better discuss prior work on tempered gradient flow dynamics and their discretizations.
- **Improve clarity in Proposition 2:** The first sentence of "Assume 2." is ambiguous and should be rephrased.
- **Rephrase for natural mathematical writing:** On page 6, the phrase "Similarly to Proposition 1" should be revised for better clarity.
- **Correct proposition numbering:** There are two instances of Proposition 1. This should be fixed.
- **Specify integral bounds:** All integrals should include their upper and lower bounds, as well as the variable of integration.
- **State all prior results:** All results from prior works used in the paper should be explicitly stated, so readers do not need to refer to other papers to understand the current one.
- **Make the structure of the paper clear**: The section on the discrete-time ULA algorithm is not mentioned in the abstract. The  connection of the section with the rest of the paper is not explicitly discussed.
- **Correct typos:**
    - In Figure 1, the plot legend $C_t$​ should be replaced with $\Sigma_t$​.
    - On page 7, `in Kuntz et al. (2023, Appendix A)` should be corrected to `by Kuntz et al. (2023, Appendix A).`
    - In the references, `Caprio et al. 2025+` should be corrected to `Caprio et al. 2025.`
    - Page 5  - `(Tempered ULA; Chehab et al. (2025))`

---

> ### Author Response · Authors · 2026-03-18
> **Rebuttal by authors**
>
> We thank the reviewer for their valuable comments and questions. We clarify the raised points below.
>
> ## Reply to ''The method of the paper is counterintuitive.''
>
> We agree that focusing on the upper bound is not sufficient.
> As lower bounds for the W flow, at least for a restricted class of targets,  already exist in the literature [1, 2], we focus on the FR case and show that
> \begin{align*}
>      \KL(\mu_t^{\textrm{FR}} || \pi)-\KL(\mu_t^{\textrm{T-FR}} || \pi)= -\int_{\int_0^te^{s-t} \lambda_s ds}^{1-e^{-t}} (1-u)\, \mathrm{Var}_{\rho_u}\left(\log\frac{\mu_0}{\pi}\right)\,du \le 0,
> \end{align*}
> which implies $\KL(\mu_t^{\textrm{T-FR}} || \pi)>\KL(\mu_t^{\textrm{FR}} || \pi)$ and allows us to conclude that the tempered FR flow is always slower than the untempered FR flow. We will add this as Proposition 3 to the revised manuscript.
>
> Additionally, we note that our results for the tempered W case differ from those in [1]. Our Proposition 1 and 4 clearly shows that the upper bound on KL decay is given by a first term corresponding to the rate of the classical W PDE, to which a bias term is added due to the fact that $\pi_t$ approaches $\pi$ only as $t\to\infty$. This decomposition is different from the one in [1], Theorem 1 and 3.  We will make this clearer in the revised manuscript.
>
> ## Mathematical details
>
> -  We will include in Appendix B.2 a complete derivation of the exact solution of the tempered FR PDE and make clear how Eq (4) can be used to derive it.
> - We will provide an explanation relating to decomposition of the WFR metric as the sum of the W and FR metrics.
> - We will include all necessary results from Chehab et al. 2025; Vempala and Wibisono, 2019 in appendix A and expand the derivations where we think they are less than clear. We will add in all references to the appendix where the proofs can be found.
> - We will clarify our derivations and improve the readability of the paper. Please let us know if you feel something other than the items in the list you provided needs clarification.
> - The bound on page (17) is obtained as follows:
>     \begin{align*}
>     \int_{\real^d} [\mu_{n-1}(x) -  \mu_{n}(x)]\log\frac{\pi(x)}{\mu_0(x)}dx &= \int \mu_{n-1}(x)(V_0(x)-V_\pi(x))dx + \int_{\real^d} \mu_{n}(x)(V_\pi(x)-V_0(x))dx\\\\
>     &\leq \int \mu_{n-1}(x)[V_0(x)-\inf_{x\in\real^d} V_\pi(x)]dx + \int_{\real^d} \mu_{n}(x)[V_\pi(x)-\inf_{x\in\real^d} V_0(x)]dx\\\\
>     &\leq\int_{\real^d} \mu_{n-1}(x) [V_0(x)-\inf_{x\in\real^d} V_0(x)]dx+\int_{\real^d} \mu_{n}(x) [V_\pi(x)-\inf_{x\in\real^d} V_\pi(x)]dx
> \end{align*}
> as $\inf_{x\in\real^d} V_\pi(x), \inf_{x\in\real^d} V_0(x)$ are constants. We will make this clear in the revision.
>
> ## Requested changes
>
> -  **Improve rigor:** See previous replies.
> - **Clarify the motivation:** In addition to providing more background information on the work in [1] and [2], we will include a section with comparisons to other tempering schedules considered in the literature, such as [3].
> - **Improve clarity in Proposition 2:** Rephrased for clarity.
> - **Rephrase for natural mathematical writing:** We now clarify that the result shows that the rate of convergence of tempered ULA is upper bounded by the rate of standard ULA plus a bias term which disappears if $\lambda_k \equiv1$.
> - **Correct proposition numbering:** Thanks for pointing this out. We will fix it in the revision.
> - **Specify integral bounds:** We will include all extrema of the integrals and the variable they are computed on.
> - **State all prior results:** We will include all necessary results from Chehab et al. 2025; Vempala and Wibisono, 2019 in appendix A and expand the derivations where we thought they were less than clear.
> - **Make the structure of the paper clear:** Thank you for pointing this out. We will update the abstract and clarify the implications of the discrete time convergence.
> - **Correct typos:** Fixed. Thank you.
>
> [1] Omar Chehab, Anna Korba, Austin J Stromme, and Adrien Vacher. Provable convergence and limitations of geometric tempering for langevin dynamics. In The Thirteenth International Conference on Learning Representations, 2025
>
> [2] Wei Guo, Molei Tao, and Yongxin Chen. Provable Benefit of Annealed Langevin Monte Carlo for Non-log-concave Sampling, 2025
>
> [3] Shirin Goshtasbpour, Victor Cohen, and Fernando Perez-Cruz. Adaptive annealed importance sampling
> with constant rate progress. In Andreas Krause, Emma Brunskill, Kyunghyun Cho, Barbara Engelhardt,
> Sivan Sabato, and Jonathan Scarlett (eds.), Proceedings of the 40th International Conference on Machine
> Learning, volume 202 of Proceedings of Machine Learning Research, pp. 11642–11658. PMLR, 23–29 Jul
> 2023.

---

### Author Response · Authors · 2026-03-26
**Rebuttal by authors**

We sincerely thank all the reviewers for careful reading and relevant remarks and questions, and positive comments.

We summarise here the main changes w.r.t. the previous version
- We clarified all mathematical derivations.
- We clarified how our results for the W flow differ/improve those in Chehab et al. (2025) and how those for the FR flow consider different dynamics than those previously considered by Mate et a. (2024) and Guo et. al (2025).
- We included Proposition 3 which shows that tempering \textbf{never} improves the FR dynamics.
- We improved section 3: our motivation for looking at the gradient flow dynamics of the tempered flows is to find the ``optimal'' $\lambda_t$ to compare with standard untempered dynamics. This point of view allows us to compare the standard untempered flows with the tempered dynamics with the optimal tempering schedule. Theoretical and empirical results show that the steepest descent schedule does not improve over the untempered dynamics.

Omar Chehab, Anna Korba, Austin J Stromme, and Adrien Vacher. Provable convergence and limitations of geometric tempering for langevin dynamics. In The Thirteenth International Conference on Learning Representations, 2025

Wei Guo, Molei Tao, and Yongxin Chen. Provable Benefit of Annealed Langevin Monte Carlo for Non-log-concave Sampling, 2025

Wei Guo, Molei Tao, Yongxin Chen.
Complexity Analysis of Normalizing Constant Estimation: from Jarzynski Equality to Annealed Importance Sampling and beyond
2025

Mate et al. Learning Interpolations between Boltzmann Densities. TMLR, 2024.

---

### Comment · Action_Editor_K6FU · 2026-04-22
**Please add metainfo to the camera ready manuscript**

The Openreview URL, dates, etc on the first page are not there as required.

---

### Decision · Action_Editor_K6FU · 2026-04-17

**Recommendation:** Accept as is

**Additional Comments:**

No additional comments.

**Audience:**

Yes

**Audience Explanation:**

Yes, this work would be of interest to community members working on sampling and tempering/annealing methods.

**Claims And Evidence:**

Yes

**Claims Explanation:**

This paper analyzes the effect of tempering on convergence for Wasserstein and Fisher-Rao flows. All reviewers agreed that claims (primarily theoretical in nature) were adequately supported by evidence.